# A human fetal liver-derived infant *MLL-AF4* acute lymphoblastic leukemia model reveals a distinct fetal gene expression program

Siobhan Rice[1], Thomas Jackson[2], Nicholas T. Crump [1], Nicholas Fordham[1], Natalina Elliott[2], Sorcha O'Byrne[2], Maria del Mar Lara Fanego[3], Dilys Addy[3], Trisevgeni Crabb[3], Carryl Dryden[3], Sarah Inglott[3], Dariusz Ladon[3], Gary Wright[3], Jack Bartram[3], Philip Ancliff[3], Adam J. Mead [1], Christina Halsey [4,5], Irene Roberts[1,2], Thomas A. Milne [1]✉ & Anindita Roy [1,2]✉

Although 90% of children with acute lymphoblastic leukemia (ALL) are now cured, the prognosis for infant-ALL remains dismal. Infant-ALL is usually caused by a single genetic hit that arises *in utero*: an *MLL/KMT2A* gene rearrangement (*MLL*-r). This is sufficient to induce a uniquely aggressive and treatment-refractory leukemia compared to older children. The reasons for disparate outcomes in patients of different ages with identical driver mutations are unknown. Using the most common *MLL*-r in infant-ALL, *MLL-AF4*, as a disease model, we show that fetal-specific gene expression programs are maintained in *MLL-AF4* infant-ALL but not in *MLL-AF4* childhood-ALL. We use CRISPR-Cas9 gene editing of primary human fetal liver hematopoietic cells to produce a t(4;11)/*MLL-AF4* translocation, which replicates the clinical features of infant-ALL and drives infant-ALL-specific and fetal-specific gene expression programs. These data support the hypothesis that fetal-specific gene expression programs cooperate with MLL-AF4 to initiate and maintain the distinct biology of infant-ALL.

[1] MRC Molecular Haematology Unit, MRC Weatherall Institute of Molecular Medicine, NIHR Oxford Biomedical Research Centre Haematology Theme, Radcliffe Department of Medicine, University of Oxford, Oxford, UK. [2] Department of Paediatrics and NIHR Oxford Biomedical Research Centre Haematology Theme, University of Oxford, Oxford, UK. [3] Department of Haematology, Great Ormond Street Hospital for Children, London, UK. [4] Wolfson Wohl Cancer Research Centre, Institute of Cancer Sciences, College of Medical, Veterinary and Life Sciences, University of Glasgow, Glasgow, UK. [5] Department of Paediatric Haematology, Royal Hospital for Children, Glasgow, UK. ✉email: thomas.milne@imm.ox.ac.uk; anindita.roy@paediatrics.ox.ac.uk

In more than 70% of infant acute lymphoblastic leukemia (infant-ALL) cases, the main driver mutation is a chromosomal translocation that leads to rearrangement of the *Mixed Lineage Leukemia* (*MLL/KMT2A*) gene (*MLL*-r), producing MLL fusion proteins[1–4]. MLL fusion proteins bind directly to gene targets where they aberrantly upregulate gene expression, partly by increasing histone-3-lysine-79 dimethylation (H3K79me2) through DOT1L recruitment[5–10]. The prevalence of *MLL*-r in infant-ALL contrasts with childhood-ALL, where *MLL*-r accounts for only 2–5% of cases[11,12]. Intriguingly, *MLL*-r childhood-ALL has better event-free survival (EFS) of 50–59%[11,13–15], compared to 19–45% in *MLL*-r infant-ALL[13,14,16]. The inferior outcome for *MLL*-r infant-ALL appears not to be due to differences in drug metabolism and/or toxicity in infants since *MLL* wild-type (*MLL*wt) infant-ALL has excellent EFS (74–93%)[1,17]. This suggests that there may be intrinsic biological differences between *MLL*-r infant-ALL and *MLL*-r childhood-ALL blasts. In support of this, the *MLL* breakpoint region tends to differ in *MLL*-r infant-ALL compared to *MLL*-r childhood-ALL[18], and infant-ALL is associated with a high frequency of the poor prognosis *HOXA*lo *MLL*-r molecular profile[19–22]. However, very little is known about the underlying reasons for these age-related differences.

A characteristic and baffling feature of *MLL*-r infant-ALL is the fact that this single oncogenic hit before birth seems to be sufficient to induce a rapidly proliferating therapy-resistant leukemia without the need for additional mutations, unlike many cases of childhood-ALL, which also originate in utero but only develop into full-blown leukemia after a second postnatal hit[23]. One reason for this could be that the specific fetal progenitors in which the translocation arises provide the permissive cellular context necessary to cooperate with *MLL*-r to induce infant-ALL[24–26].

Here, using the most common *MLL*-r infant-ALL, *MLL-AF4*, as a disease model, we identify fetal-specific gene expression programs in primary human hematopoietic cells and show that *MLL-AF4* infant-ALL, but not *MLL-AF4* childhood-ALL, maintains expression of fetal-specific genes. CRISPR-Cas9 gene editing of primary human fetal liver (FL) CD34+ cells to produce a t(4;11)/*MLL-AF4* translocation replicates the clinical features of infant-ALL in a xenograft model and drives infant-ALL- and fetal-specific molecular programs. These data support the hypothesis that developmentally regulated features of human fetal cells cooperate with MLL-AF4 to initiate and maintain the distinct biology of infant-ALL.

## Results

***MLL-AF4* infant-ALL is molecularly distinct from *MLL-AF4* childhood-ALL.** Using gene expression profiles from patients with an identical driver mutation (t(4;11)/*MLL-AF4*), we first set out to determine whether infant-ALL had a distinct molecular profile compared to childhood-ALL. It has already been shown by others that *MLL-AF4* leukemias can be divided into different subtypes based on a *HOXA*lo or *HOXA*hi gene expression profile[19–22]. In our analysis, we wanted to determine if there was a unique infant-ALL molecular signature that was independent of these subtypes.

Using a previously published patient bulk RNA-sequencing (RNA-seq) dataset[23], we carried out differential gene expression analysis between all *MLL-AF4* infant-ALL (n = 19) and *MLL-AF4* childhood-ALL (n = 5) samples. We identified 617 significantly differentially expressed genes (false discovery rate (FDR) < 0.05), 193 of which were upregulated in *MLL-AF4* infant-ALL and therefore represented an infant-ALL-specific gene expression profile (Fig. 1a, Supplementary Fig. 1a, b, Supplementary Data 1). When patients were classified as *HOXA*lo or *HOXA*hi based on

*HOXA9* expression (Supplementary Fig. 1c), we found that *HOXA*lo *MLL-AF4* infant-ALL and *HOXA*hi *MLL-AF4* infant-ALL did not separate based on these 617 genes, suggesting that this was a true infant-ALL-specific gene expression profile, irrespective of other molecular characteristics (Fig. 1a). In fact, of the top ten most significantly upregulated genes that could separate *MLL-AF4* infant-ALL from *MLL-AF4* childhood-ALL (Fig. 1b), no significant differences were observed between *HOXA*lo and *HOXA*hi *MLL-AF4* infant-ALL subsets (Fig. 1c).

**Fetal gene expression programs drive the distinct molecular profile of *MLL-AF4* infant-ALL.** We next sought to determine the extent to which normal fetal gene expression programs contribute to the distinct molecular profile of *MLL-AF4* infant-ALL. We compared bulk RNA-seq data for sorted human FL hematopoietic stem and progenitor cell (HSPC) subpopulations previously generated in our lab[27] to a human adult bone marrow (ABM) HSPC RNA-seq dataset[28]. We carried out differential gene expression analysis between comparable subpopulations of FL and ABM HSPCs along the B-lineage differentiation pathway (Fig. 1d). The hematopoietic stem cell (HSC) subpopulation showed the greatest number of differentially expressed genes between FL and ABM (3787 genes), reducing at each subsequent stage of B-lineage differentiation (1509 genes differentially expressed between FL committed B progenitors (CBPs) and ABM common lymphoid progenitors (CLPs)) (Fig. 1d). A total of 5709 genes were differentially expressed between FL and ABM in at least one HSPC subpopulation when we combined all differentially expressed gene lists (Fig. 1d and Supplementary Data 2).

We carried out clustering analysis of the patient dataset based on these 5709 genes and found that they were capable of separating *MLL-AF4* infant-ALL from *MLL-AF4* childhood-ALL (Fig. 1e and Supplementary Fig. 2a). Comparing differentially expressed genes in both the normal and leukemic setting, we found 72 genes that were significantly upregulated in at least one normal FL HSPC subpopulation and also in *MLL-AF4* infant-ALL (~40% of all genes upregulated in *MLL-AF4* infant-ALL compared to *MLL-AF4* childhood-ALL) (Supplementary Fig. 2b and Supplementary Data 2). These included some of the most significantly upregulated genes in *MLL-AF4* infant-ALL such as *HOXB4* and *IGF2BP1*, a member of the *LIN28B* gene expression pathway that has previously been reported to positively regulate *HOXB4* expression (Supplementary Fig. 2c)[29,30]. Together, these data suggest that the molecular profile of human fetal HSPCs plays a key role in determining the distinct gene expression profile of *MLL-AF4* infant-ALL.

**A CRISPR-Cas9-induced *MLL-AF4* translocation can transform human FL HSPCs in vitro.** To test the hypothesis that to accurately model *MLL-AF4* infant-ALL, the *MLL-AF4* translocation should be expressed in human fetal HSPCs, we directly induced the most common t(4;11)/*MLL-AF4* translocation in infant-ALL in primary human FL HSPCs. The translocation was induced in 13–15 postconception week (pcw) human FL CD34+ cells by CRISPR-Cas9 gene editing[31,32]. Single guide RNAs (sgRNAs) were designed to target intron 11 of *MLL* and intron 3 of *AF4* (Supplementary Fig. 3a), chosen because of their prevalence as the most common breakpoint region in infant-ALL[18]. Edited samples (n = 3; donors 1–3) (designated *CRISPR*MLL-AF4+) and biologically matched mock-edited controls (n = 3; donors 1–3) were transferred to MS-5 cocultures to facilitate the expansion of successfully edited cells along the B-lineage differentiation pathway.

By week 3 of coculture, CD19+ cell numbers were >900-fold higher in *CRISPR*MLL-AF4+ cultures compared to controls

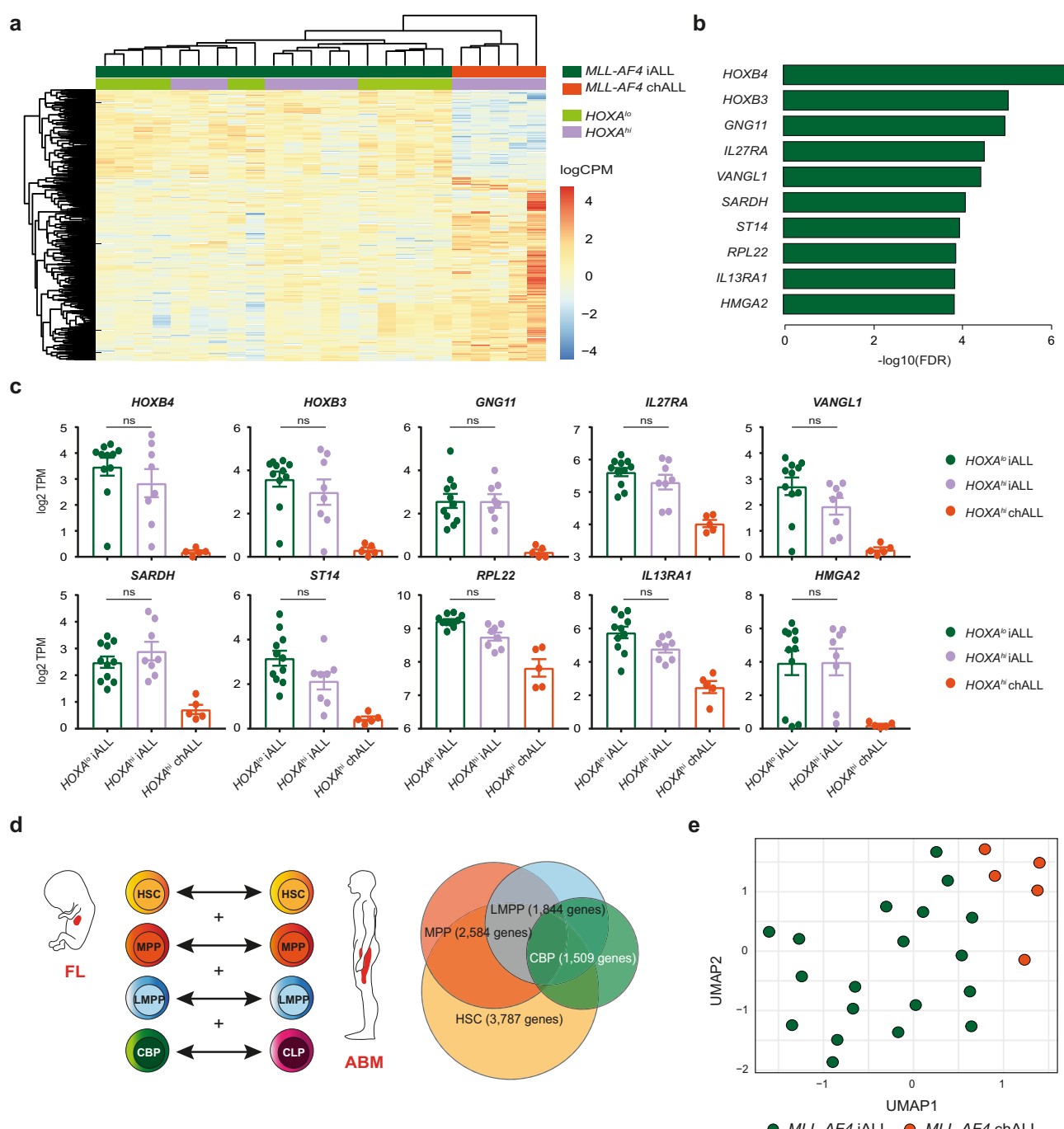

**Fig. 1 Fetal gene expression programs drive the distinct molecular profile of *MLL-AF4* infant-ALL. a** Heatmap showing clustering of *MLL-AF4* infant-ALL (iALL (dark green), $n = 19$) and *MLL-AF4* childhood-ALL (chALL (orange), $n = 5$) based on 617 significantly differentially expressed genes (FDR < 0.05, edgeR exact test; Supplementary Data 1). *HOXA*lo (light green, $n = 11$) and *HOXA*hi (purple, $n = 13$) *MLL-AF4* subsets of infant-ALL and childhood-ALL are annotated. Color scale = log 2 counts per million (logCPM). **b** Bar plot showing significance ($-\log10(FDR)$) for the ten most significantly upregulated genes in *MLL-AF4* infant-ALL. **c** Expression (log 2 transcripts per million (log 2 TPM)) of the top ten most significantly upregulated genes in *MLL-AF4* infant-ALL in *HOXA*lo *MLL-AF4* infant-ALL (iALL (light green), $n = 11$), *HOXA*hi *MLL-AF4* infant-ALL (iALL (purple), $n = 8$), and *HOXA*hi *MLL-AF4* childhood-ALL (chALL (orange), $n = 5$). Data are shown as mean ± SEM (n.s.: $p > 0.05$; one-way ANOVA with Tukey's correction for multiple comparisons). Source data are provided as a Source Data file. **d** (left) Schematic representation of differential gene expression analysis between FL and ABM. Equivalent HSPC subpopulations were compared and significantly differentially expressed genes (FDR < 0.05, edgeR exact test) for all four comparisons were combined into a master list of genes that were differentially expressed in at least one HSPC subpopulation (HSC hematopoietic stem cell, MPP multipotent progenitor cell, LMPP lymphoid-primed multipotent progenitor cell, CBP committed B progenitor, CLP common lymphoid progenitor). (right) Venn diagram showing overlap of differentially expressed genes for each HSPC subpopulation (Supplementary Data 2). **e** UMAP showing clustering of *MLL-AF4* infant-ALL (iALL (dark green), $n = 19$) and *MLL-AF4* childhood-ALL (chALL (orange), $n = 5$) from a previously published patient dataset[23] based on 5709 genes differentially expressed between FL HSPCs and ABM HSPCs (Supplementary Data 2).

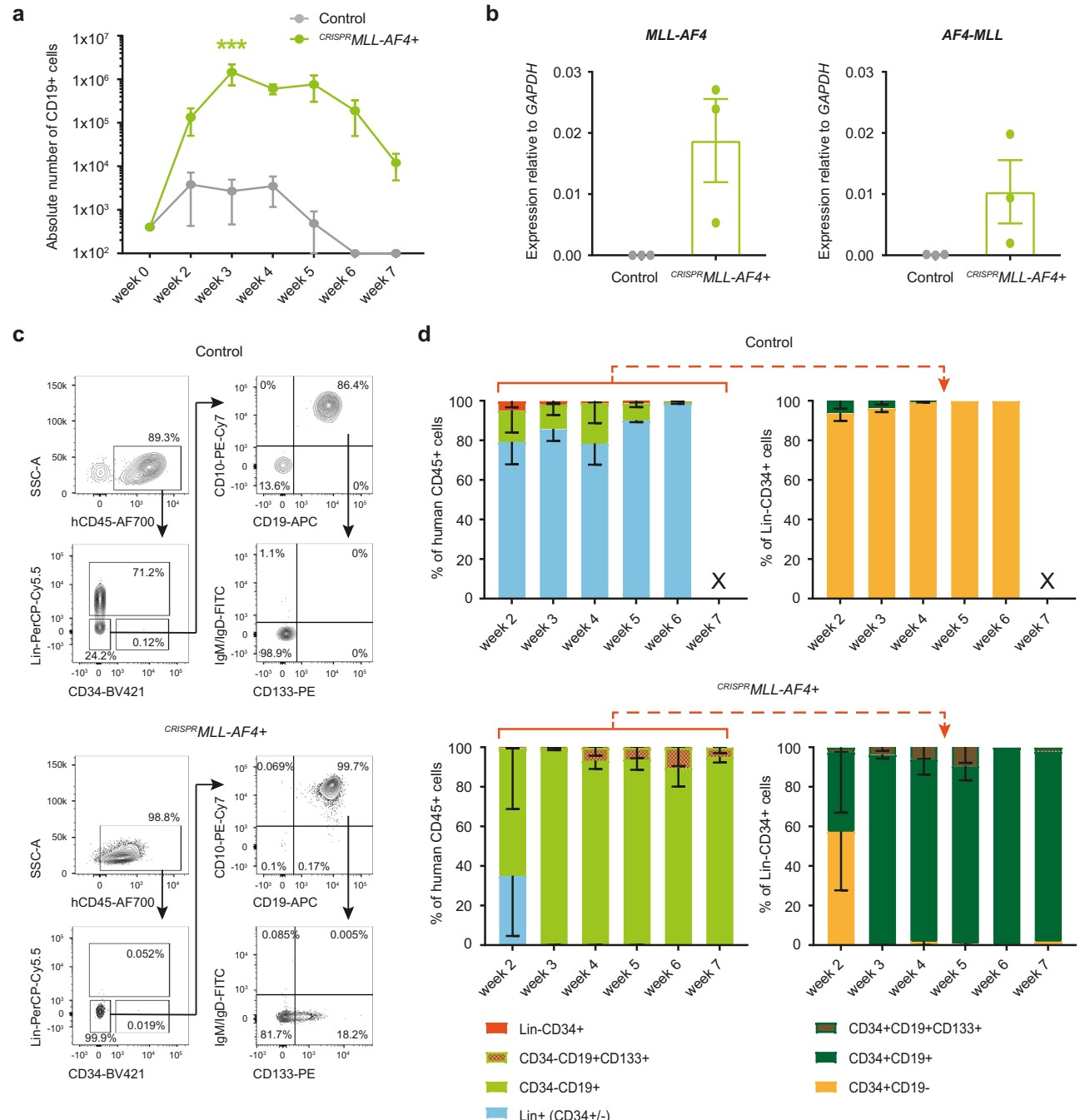

**Fig. 2 A CRISPR-Cas9-induced t(4;11)/*MLL-AF4* translocation in human FL HSPCs causes a dramatic increase in B-lineage cell proliferation in vitro.** **a** Cumulative absolute number of human CD45+CD19+ cells per well over time during *CRISPR*MLL-AF4+ and control MS-5 coculture (n = 3; donors 1–3). ***p < 0.005 (two-way ANOVA with Sidak correction for multiple comparisons). Data are shown as mean ± SEM. Source data are provided as a Source Data file. **b** RT-qPCR of human CD45+ cells showing expression of *MLL-AF4* and *AF4-MLL* (n = 3; donors 1–3) relative to *GAPDH* at week 4 of MS-5 coculture. Data are shown as mean ± SEM. Source data are provided as a Source Data file. **c** Representative flow cytometry plots of viable, single cells from control and *CRISPR*MLL-AF4+ cultures on week 4 of coculture. Custom lineage cocktail (Lin) = CD2/CD3/CD14/CD16/CD56/CD235a (Supplementary Table 4). **d** Quantification of human cell immunophenotypes as a percentage of human CD45+ (hCD45+) cells (left), and progenitor immunophenotypes as a percentage of hCD45+Lin−CD34+ cells (right), in control (n = 3; donors 1–3) and *CRISPR*MLL-AF4+ (n = 3; donors 1–3) cultures over time. X = data not shown because absolute number of hCD45+ cells is <50. Data are shown as mean ± SEM. Source data are provided as a Source Data file.

(p < 0.005), suggesting that the translocation had successfully transformed the cells (Fig. 2a). Fluorescence in situ hybridization (FISH) using *MLL* and *AF4* probes confirmed the presence of a heterozygous *MLL-AF4*/t(4;11) translocation in >80% (range 80–98%) of cells analyzed (n = 4; donors 4 and 5); Supplementary Fig. 3b. Quantitative reverse transcription-PCR (RT-qPCR)

confirmed the expression of both *MLL-AF4* and *AF4-MLL* in *CRISPR*MLL-AF4+ cultures (Fig. 2b). Virtually all human cells generated from *CRISPR*MLL-AF4+ cultures (Supplementary Fig. 3c) were CD19+ cells, compared to <20% in control cultures (Fig. 2c, d, left). Although the proportion of CD34+ cells in *CRISPR*MLL-AF4+ cultures was reduced, the absolute number of

CD34+ cells was comparable between $^{CRISPR}$MLL-AF4+ and control cultures from week 3 (Supplementary Fig. 3d). The majority of $^{CRISPR}$MLL-AF4+ CD34+ cells were CD19+ B progenitors, suggesting that *MLL-AF4*-driven B-lineage specification might occur at a progenitor stage (Fig. 2d, right). More detailed immunophenotyping showed that the majority of $^{CRISPR}$MLL-AF4+ cells were CD34−CD19+CD10+IgM/IgD− preB cells, of which ~10% aberrantly expressed the leukemia-associated marker CD133 (Fig. 2c, d), a direct gene target of the MLL-AF4 fusion protein[33]. By week 7 of coculture, when control cultures no longer produced any detectable human cells, the number of human cells in $^{CRISPR}$MLL-AF4+ cultures began to decline (Fig. 2a and Supplementary Fig. 3c), suggesting that MS-5 stroma may not be optimal for long-term maintenance of FL-derived $^{CRISPR}$MLL-AF4+ cells.

**$^{CRISPR}$MLL-AF4+ FL HSPCs cause aggressive, infant-like B-ALL in vivo.** To test whether $^{CRISPR}$MLL-AF4+ cells could generate leukemia in vivo, human FL CD34+ cells (13 pcw, $n = 4$; donors 1, 2, 3, and 6) were edited as before and transplanted into sublethally irradiated NOD.Cg-Prkdc$^{scid}$Il2rg$^{tm1Wjl}$/SzJ (NSG) mice ($^{CRISPR}$MLL-AF4+, $n = 3$: donors 1 and 2; control, $n = 5$: donors 1, 2, 3, and 6). By 12 weeks posttransplant, human CD45+ cells were detected in peripheral blood (PB) (Supplementary Fig. 4a) and RT-qPCR showed that both *MLL-AF4* and *AF4-MLL* fusion transcripts were clearly detectable in human CD45+ cells from $^{CRISPR}$MLL-AF4+ mice (Supplementary Fig. 4b). B-ALL rapidly developed in all three $^{CRISPR}$MLL-AF4+ mice with a median latency of 18 weeks, whereas no control mice (0/5) developed any form of leukemia (Fig. 3a). FISH analysis (Fig. 3b and Supplementary Fig. 4c) and Sanger sequencing (Supplementary Table 1 and Supplementary Fig. 4d) confirmed the presence of a heterozygous *MLL-AF4*/t(4;11) translocation in $^{CRISPR}$MLL-AF4+ cells. Greater than 87% of cells were positive for *MLL-AF4* by FISH (range 87–99%, $n = 3$) (Supplementary Fig. 4c). In addition, Sanger sequencing and FISH analysis confirmed that the wt allele of *MLL* had neither gained indels that would affect its expression (Supplementary Fig. 4e and Supplementary Table 1) nor translocated to any of the four most common *MLL* fusion partners other than *AF4* (*AF6*, *AF9*, *AF10*, and *ENL*) (Supplementary Fig. 4f), and karyotyping confirmed that no other major structural abnormalities had been caused post-CRISPR-Cas9 editing (Supplementary Fig. 4g and Supplementary Table 2). Sanger sequencing of potential off-target editing sites[31,32] predicted in silico showed that, in the $^{CRISPR}$MLL-AF4+ clones that grew out in primary recipient mice ($n = 3$; donors 1 and 2), no indels were present at these loci (Supplementary Table 1).

The B-ALL in $^{CRISPR}$MLL-AF4+ mice recapitulated key phenotypic features of infant-ALL, including circulating blasts in the PB (Supplementary Fig. 4h) and blast infiltration into the spleen and liver (Fig. 3c and Supplementary Fig. 4h). $^{CRISPR}$MLL-AF4+ mice also had central nervous system (CNS) disease, with extensive parameningeal blast cell infiltration (Fig. 3d); a key clinical feature of infant-ALL. As chemo-resistance is also an important feature of *MLL*-r infant-ALL, we compared the responses of $^{CRISPR}$MLL-AF4+ ALL blasts to prednisolone and L-asparaginase with ALL patient-derived xenograft (PDX) samples and the SEM and KOPN8 *MLL*-r cell lines and found that $^{CRISPR}$MLL-AF4+ blasts showed similar levels of in vitro drug resistance to previous reports of treatment-resistant patient samples[34–36] (Supplementary Fig. 4i and Supplementary Table 2). Secondary ($n = 4$; donors 1 and 2) and tertiary ($n = 3$; donors 1 and 2) recipient mice all developed B-ALL with significantly reduced latency compared to primary recipients (median survival

11.5 weeks in secondary ($p < 0.02$) and 8 weeks in tertiary ($p < 0.03$)) (Fig. 3a).

Although the clinicopathological features were the same in all leukemic mice (Supplementary Table 2), 2/3 $^{CRISPR}$MLL-AF4+ mice had a CD19+CD10−CD20−IgM/IgD−CD34+/− proB ALL immunophenotype (Fig. 3e and Supplementary Fig. 5a), while the remaining mouse had a preB ALL immunophenotype, with the majority of cells being CD19+CD10+CD20−IgM/IgD−CD34− (Supplementary Fig. 5a, b). Further characterization of $^{CRISPR}$MLL-AF4+ proB ALL (the most common type seen in patients) revealed that it recapitulated the immunophenotype of *MLL-AF4* infant-ALL, including heterogeneous expression of CD133[33], NG2[37], and CD24 (Fig. 3e and Supplementary Fig. 5c). Sequencing of the IgH locus showed that $^{CRISPR}$MLL-AF4+ ALL was clonal (Supplementary Table 2). The proportion of blasts that were CD34+ did not correlate with CD10 expression (Supplementary Fig. 5a and Supplementary Table 2), nor did it increase significantly in secondary recipients (Supplementary Fig. 5d and Supplementary Table 2). This is in keeping with data from primary *MLL*-r patient samples, where CD34 expression is known to be heterogeneous (Supplementary Fig. 5d). In fact, all clinicopathological and immunophenotypic features of primary $^{CRISPR}$MLL-AF4+ ALL were maintained in secondary recipients, including CNS disease (Supplementary Table 2). Together, these data show that a CRISPR-Cas9-induced t(4;11)/*MLL-AF4* translocation in human FL HSPCs is sufficient to promote a rapidly progressive, fatal, transplantable B-ALL that recapitulates key features of infant-ALL.

**$^{CRISPR}$MLL-AF4+ ALL recapitulates the molecular profile of *MLL-AF4* ALL in patients.** To characterize the transcriptomic changes underlying leukemic transformation, human CD19+ cells were sorted from the BM of primary $^{CRISPR}$MLL-AF4+ ($n = 3$; donors 1 and 2) and control ($n = 3$; donors 1 and 2) mice for bulk RNA-seq (Supplementary Fig. 6a). Among the 1068 differentially expressed genes between $^{CRISPR}$MLL-AF4+ ALL and control CD19+ cells were many genes known to be upregulated in *MLL-AF4* ALL, including *FLT3*, *MEIS1*, and *RUNX1* (Supplementary Fig. 6b). We compared bulk RNA-seq from control and $^{CRISPR}$MLL-AF4+ BM to two independent patient datasets[23,27] and found that, on a transcriptome-wide level, $^{CRISPR}$MLL-AF4+ ALL more closely resembled *MLL-AF4* ALL patients compared to *MLL*wt ALL patients (Fig. 4a, b and Supplementary Fig. 6c). In addition, we carried out differential gene expression analysis between *HOXA*$^{lo}$ *MLL-AF4* infant-ALL, *HOXA*$^{hi}$ *MLL-AF4* infant-ALL, and *HOXA*$^{hi}$ *MLL-AF4* childhood-ALL patients to derive a list of 765 genes that separate these patient subsets (Supplementary Data 3). Based on these genes, we found that all $^{CRISPR}$MLL-AF4+ ALLs clustered with *HOXA*$^{lo}$ *MLL-AF4* infant-ALL, suggesting that they all represent *HOXA*$^{lo}$ *MLL-AF4* infant-ALL, regardless of their immunophenotype (Fig. 4c). Interestingly, while all $^{CRISPR}$MLL-AF4+ mice showed negligible *HOXA9* expression characteristic of this molecular profile, the expression pattern of *IRX1*, a gene that is usually overexpressed in *HOXA*$^{lo}$ ALL[20,21] was variable. This finding was similar to *HOXA*$^{lo}$ *MLL-AF4* infant-ALL patients (Fig. 4d).

By chromatin immunoprecipitation-seq (ChIP-seq), we observed a clear genome-wide correlation between the MLL-AF4-binding profile in $^{CRISPR}$MLL-AF4+ ALL, the *MLL-AF4* B-ALL SEM cell line[38], and a primary *MLL-AF4* ALL patient sample[39]. In addition, we observed a substantial overlap in MLL-AF4 target genes (2323 genes bound by MLL-AF4 in all three datasets) (Fig. 4e, Supplementary Fig. 6d, and Supplementary

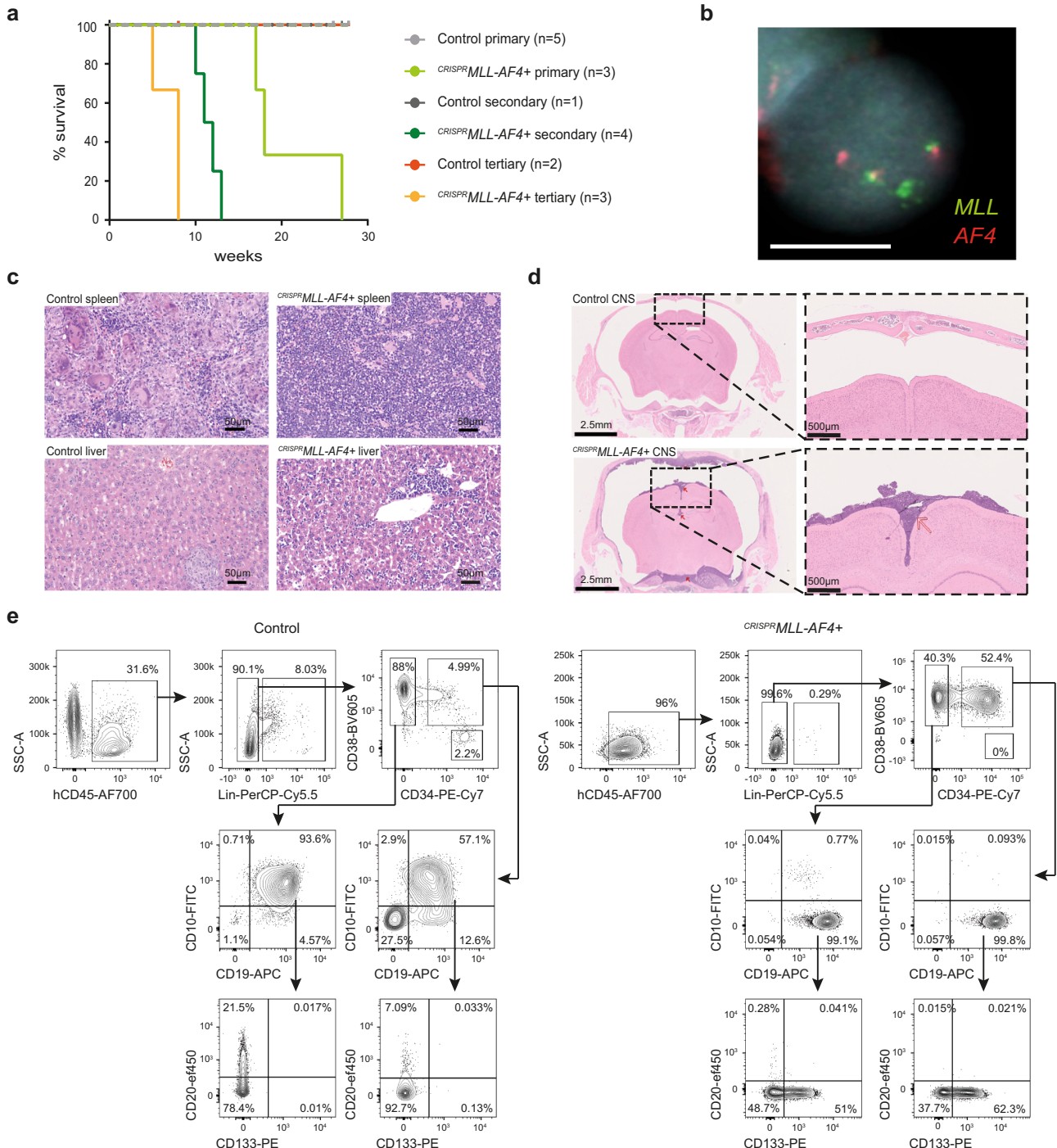

**Fig. 3 $^{CRISPR}$MLL-AF4+ cells give rise to a B-ALL in vivo that recapitulates many key features of infant-ALL. a** Leukemia-free survival for primary ($^{CRISPR}$MLL-AF4+ $n = 3$: donors 1 and 2; control $n = 5$: donors 1, 2, 3, and 6), secondary ($^{CRISPR}$MLL-AF4+ $n = 4$: donors 1 and 2; control $n = 1$: donor 1) and tertiary ($^{CRISPR}$MLL-AF4+ $n = 3$: donors 1 and 2; control $n = 2$: donor 1) recipient mice. Mice culled with no signs of leukemia (see "Methods") are censored (shown as a tick above the line). Latency significantly reduced for secondary ($p = 0.018$) and tertiary ($p = 0.03$) $^{CRISPR}$MLL-AF4+ compared to primary $^{CRISPR}$MLL-AF4+ (log-rank (Mantel–Cox) test). Source data are provided as a Source Data file. **b** Dual FISH (MLL probe = green; AF4 probe = red) showing heterozygous chromosomal translocation in a $^{CRISPR}$MLL-AF4+ cell isolated from the spleen of a primary recipient mouse. Representative image of 200 cells analyzed ($n = 3$; donor 1–2). **c** Representative H&E staining of spleen and liver from control and $^{CRISPR}$MLL-AF4+ primary recipient mice ($n = 3$; donor 1–3). Scale bar = 50 μm. **d** Representative H&E staining of control and $^{CRISPR}$MLL-AF4+ primary recipient mouse heads (scale bar = 2.5 mm; red arrows = regions of concentrated blast cell infiltration). High magnification images (scale bar = 500 μm) highlight striking parameningeal blast cell infiltration in $^{CRISPR}$MLL-AF4+ mice (red arrow) but not control ($n = 4$; donor 1–2 primary and secondary recipients). **e** Representative flow cytometry plots of viable, single cells in control and $^{CRISPR}$MLL-AF4+ BM at termination (weeks 17 and 18, respectively).

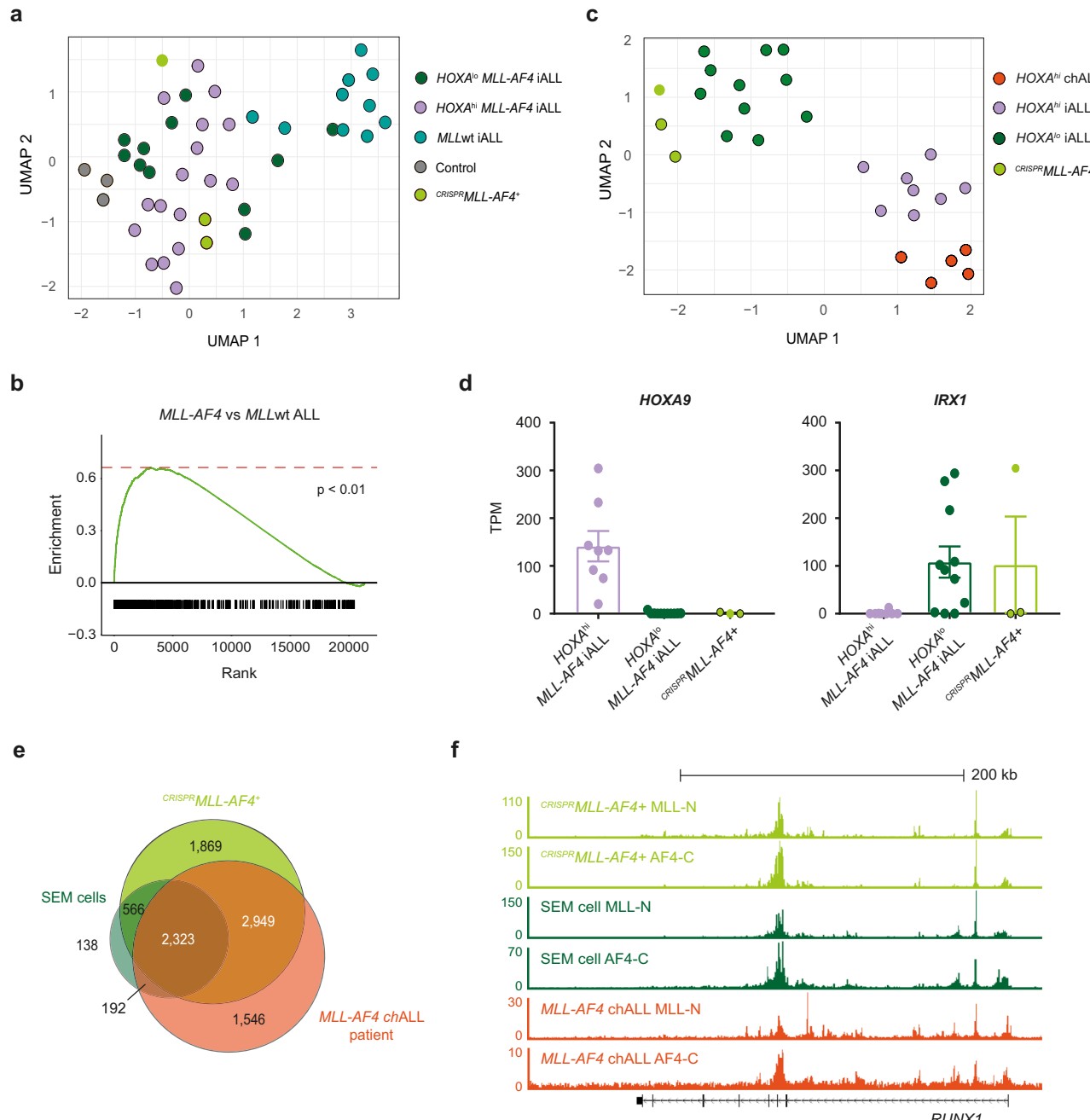

**Fig. 4 $^{CRISPR}$MLL-AF4+ ALL recapitulates the molecular profile of MLL-AF4 ALL in patients. a** UMAP showing clustering of CD19+ cells from $^{CRISPR}$MLL-AF4+ (light green; black border = proB ALL, no border = preB-ALL) and control (gray) mice with HOXA$^{lo}$ MLL-AF4 (dark green, $n = 11$), HOXA$^{hi}$ MLL-AF4 (purple, $n = 20$), and MLLwt (blue, $n = 9$) infant-ALL patient samples from a publicly available dataset[27] based on 7041 significantly differentially expressed genes (FDR < 0.05, edgeR glm test) between $^{CRISPR}$MLL-AF4+ ALL, controls, MLL-AF4 infant-ALL, and MLLwt infant-ALL. **b** Gene set enrichment analysis (GSEA)[52] showing $^{CRISPR}$MLL-AF4+ ALL is more enriched for genes that are upregulated in MLL-AF4 ALL compared to MLLwt ALL (1000 genes) when compared to CD19+ cells from control mice ($p < 0.01$). **c** UMAP showing clustering of $^{CRISPR}$MLL-AF4+ (light green; black border = proB ALL, no border = preB-ALL) with HOXA$^{lo}$ MLL-AF4 (dark green, $n = 11$) HOXA$^{hi}$ MLL-AF4 infant-ALL (purple, $n = 8$), and HOXA$^{hi}$ MLL-AF4 childhood-ALL (orange, $n = 5$) from a publicly available patient dataset[23] based on 765 significantly differentially expressed genes (FDR < 0.05; edgeR glm test) between these three patient subsets (Supplementary Data 3). **d** Expression (TPM) of HOXA9 and IRX1 in HOXA$^{hi}$ MLL-AF4 infant-ALL (purple, $n = 8$), HOXA$^{lo}$ MLL-AF4 infant-ALL (dark green, $n = 11$) and $^{CRISPR}$MLL-AF4+ ALL (light green; black border = proB ALL, no border = preB ALL). Data are shown as mean ± SEM. Source data are provided as a Source Data file. **e** Venn diagram showing overlap of MLL-AF4-bound genes (genes with an MLL-AF4 peak in the gene body) in $^{CRISPR}$MLL-AF4+ ALL in a primary recipient mouse, the MLL-AF4+ SEM cell line and a primary MLL-AF4 childhood-ALL (chALL) patient sample. MLL-AF4 peaks = directly overlapping MLL-N and AF4-C peaks. **f** Representative ChIP-seq tracks at the MLL-AF4 target gene, RUNX1 in $^{CRISPR}$MLL-AF4 + ALL, the SEM cell line and a primary MLL-AF4 childhood-ALL (chALL) patient sample.

Data 4) with strikingly similar binding profiles within target genes, such as *RUNX1* (Fig. 4f).

**FL-derived ^CRISPR^*MLL-AF4*+ ALL specifically recapitulates *MLL-AF4* infant-ALL.** Finally, we wanted to ask whether inducing an *MLL-AF4* translocation in human FL HSPCs gave rise to a model that specifically recapitulated the molecular profile of *MLL-AF4* infant-ALL. The only humanized mouse model of *MLL-AF4* ALL that has previously been published introduced a chimeric *MLL-Af4* fusion gene into human cord blood (CB) HSPCs (hereafter referred to as CB *MLL-Af4*+ ALL)[40]. While we acknowledge the fact that it was generated using a different technique and did not harbor the reciprocal *AF4-MLL* translocation, we hypothesized that this model may represent a neonatally derived (non-fetal) ALL to which our model could be compared.

To examine the fetal and postnatal gene expression programs that are key to determining the age-related differences between *MLL-AF4* ALLs, we used the 139 genes up- or downregulated in both FL (compared to ABM) and *MLL-AF4* infant-ALL (compared to *MLL-AF4* childhood-ALL) (Supplementary Data 2). Clustering analysis based on this core fetal-specific infant-ALL gene list showed that ^CRISPR^*MLL-AF4*+ ALL clustered with *MLL-AF4* infant-ALL, whereas both *MLL-AF4* childhood-ALL and CB *MLL-Af4*+ ALL formed their own, separate clusters (Fig. 5a). To explore this in more detail, we carried out differential gene expression analysis between ^CRISPR^*MLL-AF4*+ ALL and CB *MLL-Af4*+ ALL followed by gene set enrichment analysis (GSEA). We found that ^CRISPR^*MLL-AF4*+ ALL was significantly enriched for genes upregulated in both *MLL-AF4* infant-ALL ($p < 0.03$) and FL HSPCs ($p < 0.001$) compared to CB *MLL-Af4*+ ALL (Fig. 5b).

Comparing MLL-AF4 binding at promoters genome-wide in both models, we found that MLL-AF4 in ^CRISPR^*MLL-AF4*+ ALL showed greater enrichment (normalized ChIP-seq reads/bp) at the promoters of infant-ALL- and FL-specific genes compared to MLL-Af4 in CB *MLL-Af4*+ ALL. However, at all other genes, MLL-AF4/MLL-Af4 enrichment was comparable (Fig. 5c). At infant-ALL- and FL-specific genes *IGF2BP1* (Fig. 5d) and *HOXB4* (Fig. 5e), we observed an MLL-AF4 peak at the promoter in ^CRISPR^*MLL-AF4*+ ALL but not CB *MLL-Af4*+ ALL. These data suggested that MLL-AF4 may play an active role in maintaining fetal gene expression programs in infant-ALL.

Increased levels of H3K79me2 are a commonly used marker of MLL-AF4 activity[5,6,33,41]. Therefore, using one of the unique features of our model, we carried out H3K79me2 ChIP-seq in identical primary human FL HSPCs before and after leukemic transformation. We observed increased levels of H3K79me2 in infant-ALL- and FL-specific genes such as *IGF2BP1* (Fig. 5d) and *HOXB4* (Fig. 5e) in ^CRISPR^*MLL-AF4*+ ALL, further suggesting that MLL-AF4 actively maintains the expression of these fetal-specific genes in *MLL-AF4* infant-ALL.

## Discussion

The mechanisms by which the same *MLL*-r driver mutation could cause more aggressive disease and worse outcomes in infant-ALL compared to childhood-ALL have always been unclear. We hypothesized that there must be intrinsic biological differences between infant-ALL and childhood-ALL blasts, unrelated to their shared driver mutation, that underlie these age-related differences. To explore this, we used the most common *MLL*-r infant-ALL, *MLL-AF4*, as a disease model and set out to identify age-related differences on the transcriptomic level. Using primary patient data[23], we have identified the unique molecular profile of *MLL-AF4* infant-ALL. Importantly, we find that this profile is

present regardless of other well-studied molecular characteristics of the ALL, such as *HOXA* status[19-22]. Reasoning that this profile drives the distinct phenotype of infant-ALL, we then set out to identify factors that could explain it. We find that maintenance of fetal-specific gene expression programs accounts for a large proportion (~40%) of the unique molecular profile of *MLL-AF4* infant-ALL, suggesting that it is the fetal target cells in which the translocation arises that provide the permissive cellular context for aggressive infant-ALL.

Human fetal HSPCs are more proliferative than ABM HSPCs[42,43] and they differentiate down distinct developmental pathways[44,45], some of which are virtually absent in postnatal life. Therefore, maintenance of fetal HSPC characteristics provides a possible explanation for the highly proliferative, therapy-resistant nature of infant-ALL. However, one of the biggest challenges to understanding the biology of infant-ALL has been the lack of an appropriate model that captures the unique characteristics and aggressive nature of the disease. By targeting a t(4;11)/*MLL-AF4* translocation to primary human FL HSPCs, we have created a faithful humanized *MLL-AF4* infant-ALL model. Our results confirm that a human fetal cell context is permissive to give rise to an ALL that recapitulates key phenotypic and molecular features of poor prognosis *MLL-AF4* infant-ALL.

We targeted the t(4;11)/*MLL-AF4* translocation to CD34+ FL cells, which represent a mixture of different HSPC types. The immunophenotypic heterogeneity we observed among primary ^CRISPR^*MLL-AF4*+ mice, with 2/3 showing a proB and 1/3 showing a preB immunophenotype, may be a consequence of the translocation occurring in different progenitor cell types. Interestingly, however, no other significant differences were observed between proB and preB ^CRISPR^*MLL-AF4*+ ALL. First, no clinicopathological differences were observed, which may suggest that it is the shared fetal characteristics, more so than a cell-type-specific context, that drive the aggressive phenotypic features of infant-ALL, such as treatment resistance and CNS disease. Second, all ^CRISPR^*MLL-AF4*+ ALLs represented the *HOXA*^lo subset of *MLL-AF4* infant-ALL. While this may draw an interesting parallel with the higher frequency of the *HOXA*^lo subset observed in *MLL*-r infant-ALL patients[19-22], we cannot draw conclusions from these data about the specific cell of origin of infant-ALL and/or the drivers of the *HOXA*^lo/*HOXA*^hi molecular profiles. It will be interesting in the future to target the t(4;11)/*MLL-AF4* translocation to specific fetal HSPC subsets to ask whether leukemic transformation and *HOXA* status is determined by gestational age, hematopoietic site, or progenitor cell type. Finally, all ^CRISPR^*MLL-AF4*+ ALLs showed expression of both reciprocal fusion genes, *MLL-AF4* and *AF4-MLL*. The contribution of *AF4-MLL* to the initiation of *MLL-AF4* leukemia has been a topic of debate in the field[46,47]. However, our editing approach has not allowed us to address the question of the relative importance of *AF4-MLL* to transformation. In the future, our editing approach could potentially be adapted to express only one or both of the reciprocal fusion genes.

As well as providing insights into MLL-AF4 function in a human fetal cell context, ^CRISPR^*MLL-AF4*+ ALL provides a preclinical model for translational studies that specifically recapitulates poor prognosis infant-ALL. For example, the CNS disease observed in ^CRISPR^*MLL-AF4*+ ALL is a common clinical feature of infant-ALL that can lead to CNS relapse in these patients[48]. Therefore, the ability of novel treatments to eradicate blasts from the CNS is an important consideration, and this can be tested in ^CRISPR^*MLL-AF4*+ ALL. Going forward, the unique, age-related molecular profile of *MLL-AF4* infant-ALL defined here can be mined for potential novel targets to specifically treat poor prognosis infant-ALL, and these novel treatments can then be tested in ^CRISPR^*MLL-AF4*+ ALL.

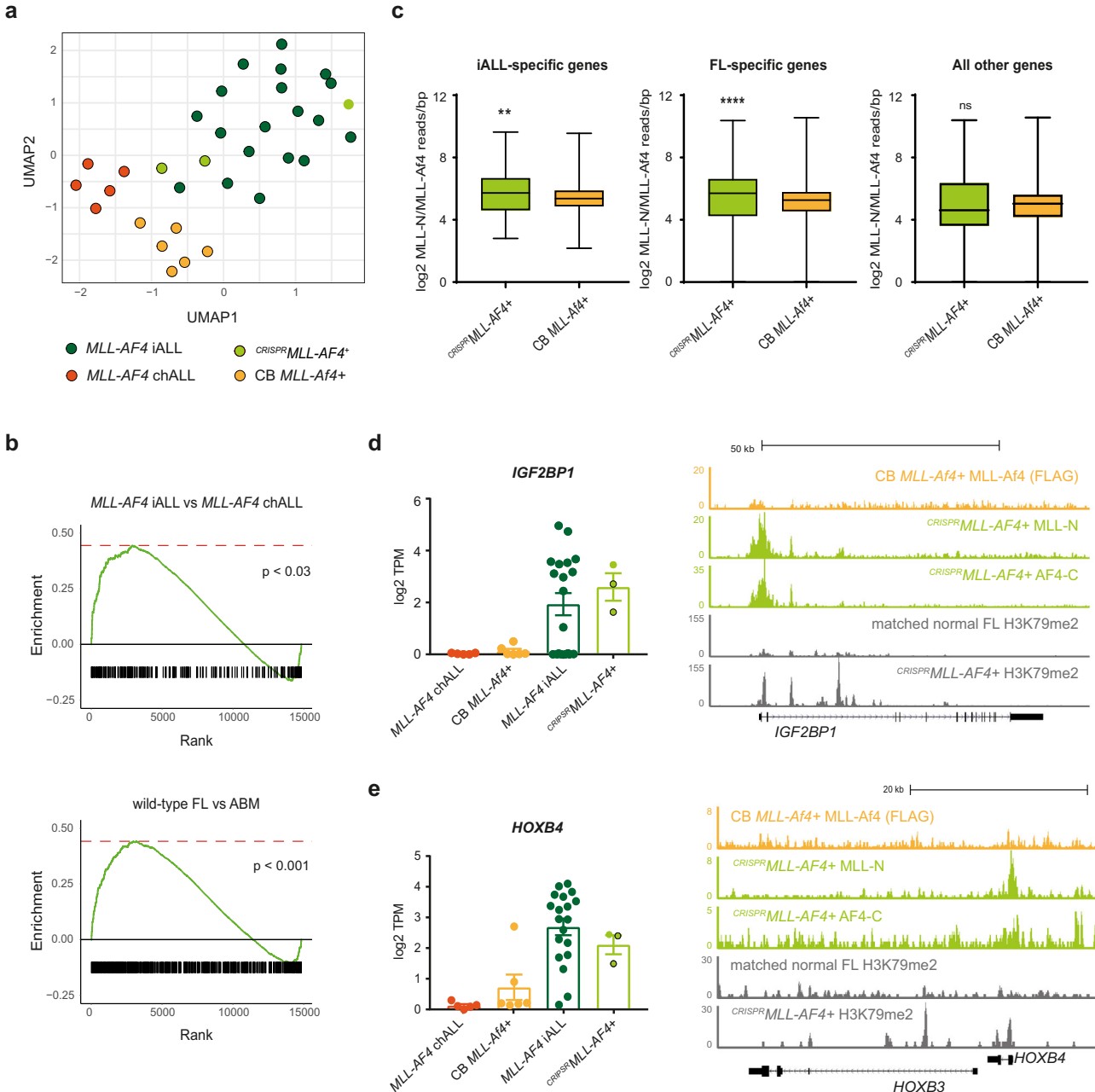

**Fig. 5 FL-derived $^{CRISPR}$MLL-AF4+ ALL specifically recapitulates the molecular profile of *MLL-AF4* infant-ALL. a** UMAP showing clustering of $^{CRISPR}$MLL-AF4+ ALL (light green; black border = proB ALL, no border = preB ALL), *MLL-AF4* infant-ALL (iALL (dark green), n = 19), *MLL-AF4* childhood-ALL (chALL (orange), n = 5)[20], and CB *MLL-Af4+* ALL[35] (yellow, n = 6) based on a set of genes differentially expressed in both FL (compared to ABM) and *MLL-AF4* infant-ALL (compared to *MLL-AF4* childhood-ALL) (139 genes; Supplementary Data 2). **b** Gene set enrichment analyses (GSEA)[52] showing $^{CRISPR}$MLL-AF4+ ALL is significantly more enriched for genes upregulated in *MLL-AF4* infant-ALL (compared to *MLL-AF4* childhood-ALL) (617 genes, p < 0.03) and in FL (compared to ABM) (5709 genes, p < 0.001) when compared to CB *MLL-Af4+* ALL. **c** Box-and-whisker plots showing MLL-AF4 enrichment (MLL-N ChIP-seq reads/bp normalized to 10^7 total reads) for $^{CRISPR}$MLL-AF4+ ALL and CB *MLL-Af4+* ALL (FLAG ChIP-seq) at infant-ALL-specific genes (193 genes, two-tailed Mann–Whitney test, p = 0.012) and FL-specific genes (3949 genes, two-tailed Mann–Whitney test, p < 0.0001). Data are shown as mean ± SEM. Source data are provided as a Source Data file. **d** (left) Barplot showing expression of *IGF2BP1* in *MLL-AF4* childhood-ALL (chALL (orange), n = 5)[20], CB *MLL-Af4+* ALL (yellow, n = 6)[35], *MLL-AF4* infant-ALL (iALL (dark green), n = 19)[20] and $^{CRISPR}$MLL-AF4 + ALL (light green; black border = proB ALL, no border = preB ALL). Data shown as mean ± SEM. Source data are provided as a Source Data file. (Right) MLL-Af4 (FLAG ChIP-seq) in CB *MLL-Af4+* ALL, MLL-N, and AF4-C ChIP-seq in $^{CRISPR}$MLL-AF4+ ALL and H3K79me2 ChIP-seq in $^{CRISPR}$MLL-AF4+ ALL and matched unedited FL sample at *IGF2BP1*. ChIP-seq data shown are normalized to 10^7 total reads. **e** (left) Barplot showing expression of *HOXB4* in *MLL-AF4* childhood-ALL (chALL (orange), n = 5)[20], CB *MLL-Af4+* ALL (yellow, n = 6)[35], *MLL-AF4* infant-ALL (iALL (dark green), n = 19)[20], and $^{CRISPR}$MLL-AF4+ ALL (light green; black border = proB ALL, no border = preB ALL). Data are shown as mean ± SEM. Source data are provided as a Source Data file. (Right) MLL-Af4 (FLAG ChIP-seq) in CB *MLL-Af4+* ALL, MLL-N, and AF4-C ChIP-seq in $^{CRISPR}$MLL-AF4+ ALL and H3K79me2 ChIP-seq in $^{CRISPR}$MLL-AF4+ ALL and matched unedited FL sample at *HOXB3/HOXB4*. ChIP-seq data shown are normalized to 10^7 total reads.

## Methods

**Samples**. Donated fetal tissue was provided for purposes of this research by the Human Developmental Biology Resource (www.hdbr.org), regulated by the UK Human Tissue Authority (www.hta.gov.uk) and covered under ethics granted by NHS Health Research Authorities: North East—Newcastle & North Tyneside Research Ethics Committee (REC: 18/NE/0290) and London—Fulham Research Ethics Committee (18/LO/0822). Informed consent was obtained from all participants, who donated human fetal tissue for research without receiving any monetary compensation. FL samples used for CRISPR/Cas9 *MLL-AF4* translocation experiments underwent CD34 magnetic bead selection at the time of sample processing and were cryopreserved for future use as described previously[49]. ALL patient samples were obtained from Blood Cancer UK Childhood Leukemia Cell Bank, UK after appropriate review of our research project to ensure that it was covered under their ethics approval granted by NHS HRA South West - Central Bristol Research Ethics Committee (REC: 16/SW/0219). Informed consent was obtained from all participants or those with parental responsibility, and participants did not receive any monetary compensation. Infant and pediatric *MLL*-r ALL patients being treated at Great Ormond Street Hospital for Children, London had immunophenotypic analysis performed as part of their diagnostic workup after informed consent was obtained from all participants or those with parental responsibility. All patient samples/data were anonymized at source, assigned a unique study number, and linked. *MLL*-r and *ETV6-RUNX1*+ ALL PDX cells were provided by the Halsey lab.

**Animals**. All experiments were performed under a project license approved by the UK Home Office under the Animal (Scientific Procedures) Act 1986 after approval by the Oxford Clinical Medicine Animal Welfare and Ethical Review Body; and in accordance with the principles of 3Rs (replacement, reduction, and refinement) in animal research. All experimental animals were 8–12-week-old female NSG mice ($n = 18$). Mice were housed in individually ventilated cages, and kept at a 12-h light/dark cycle, 21–22 °C temperature, and 45–65% relative humidity. They had red tunnels or houses and balconies in the cages as enrichment.

**CRISPR-Cas9 *MLL-AF4* translocation**. CRISPR-Cas9 genome editing was carried out using a previously described protocol[50]. *MLL* and *AF4* sgRNAs (Synthego; Supplementary Table 3) were first tested for editing efficiency individually in FL CD34+ cells. Cryopreserved CD34+ cells from a single primary human FL sample were thawed and placed into suspension culture at a density of $2.5 \times 10^5$ cells/ml in StemLine II (Sigma) supplemented with stem cell factor (SCF) (100 ng/ml), FLT-3-ligand (FLT3L) (100 ng/ml), and thrombopoietin (100 ng/ml) (Peprotech) for 12 h. Cells were harvested and electroporated with either (i) Cas9 protein (IDT) only or (ii) a Cas9/sgRNA ribonucleoprotein (RNP) using a Neon™ Transfection System (Thermo Fisher). Electroporated cells were placed into fresh suspension culture media to recover overnight. Cells were harvested and bulk genomic DNA was extracted using a DNeasy Blood and Tissue Kit (Qiagen). An ~1 kb region of DNA around the target cut site was amplified by PCR and Sanger sequenced (Eurofins). Sanger sequencing traces from samples edited with RNPs were compared to traces from Cas9-only controls using the ICE Analysis online tool (Synthego, https://ice.synthego.com). Editing efficiency is reported as the percentage of indels detected (Supplementary Fig. 3a).

For each CRISPR-Cas9 *MLL-AF4* translocation experiment, cryopreserved CD34+ cells from a single 13–15 pcw primary human FL underwent suspension culture as described. Cells were harvested and electroporated with either (i) Cas9 protein (IDT) only, (ii) Cas9 protein plus *MLL*-sgRNA only, as biologically matched controls, or (iii) a 1:1 mix of Cas9/MLL-sgRNA and Cas9/AF4-sgRNA RNPs using a Neon™ Transfection System (Thermo Fisher). Electroporated cells were placed into fresh suspension culture media to recover overnight before subsequent in vitro culture and in vivo transplantation experiments.

**MS-5 stroma coculture**. Electroporated FL CD34+ cells ($^{CRISPR}$MLL-AF4+ and control) were plated onto a confluent layer of MS-5 stromal cells in a 24-well plate at a density of 2000 cells/well in αMEM (Gibco) supplemented with 10% heat-inactivated batch-tested fetal bovine serum (FBS), 100 U/ml penicillin, 100 µg/ml streptomycin, 2 mM L-glutamine, 50 µM 2-mercaptoethanol, 10 mM HEPES, SCF (20 ng/ml), FLT3L (10 ng/ml), interleukin-2 (IL-2) (10 ng/ml), and IL-7 (5 ng/ml) (Peprotech). Cultures were maintained as described previously[45,49]. Cells were harvested for flow cytometry analysis once a week beginning at week 2 of culture. *MLL-AF4* and *AF4-MLL* RT-qPCRs were carried out on week 4 of culture.

**Xenograft transplantation**. The 8–12-week-old female NSG mice were sublethally irradiated with two doses of 1.25 Gy 6 h apart (2.5 Gy total) and injected via the tail vein with 25,000–35,000 edited FL cells ($^{CRISPR}$MLL-AF4+, $n = 3$; Cas9 control, $n = 5$; or Cas9 plus *MLL*-sgRNA control, $n = 1$) plus 30,000 wt, unedited, sex-mismatched FL CD34+ carrier cells. Engraftment was monitored by PB sampling every 3 weeks. Human CD45+ cells were sorted from PB samples to carry out *MLL-AF4* and *AF4-MLL* RT-qPCR for the detection of successfully edited cells. Animals were monitored regularly using a standardized physical scoring system, and any mouse found to be in distress was humanely killed. Mice were considered leukemic if they met at least three of the following criteria: (i) overt signs of disease

(hunching, lack of movement, weight loss, paralysis), (ii) splenomegaly, (iii) PB blast count over 50%, (iv) peripheral organ infiltration, and (v) detection of the *MLL-AF4* translocation in both BM and spleen.

**Flow cytometry**. Cells were stained with fluorophore-conjugated monoclonal antibodies in phosphate-buffered saline with 2% FBS and 1 mM EDTA for 30 min and analyzed using BD LSR II or Fortessa X50 instruments using BD FACSDiva software (v8.0.2). Antibodies used are detailed in Supplementary Table 4. Flow cytometry antibodies were validated by titration in-house using primary human fetal mononuclear cells or NSG mouse BM. Analysis was performed using FlowJo software (v10.7.1), where gates were set using unstained and fluorescence minus one control.

**Histopathology**. On termination, samples of ~0.5–1 cm$^2$ were taken from the spleen and liver of $^{CRISPR}$MLL-AF4+ and Cas9 control mice and fixed in 10% formaldehyde. After fixation, tissues were processed and paraffin-embedded. Four micrometers of paraffin sections were cut onto Superfrost Plus adhesive slides (VWR, Cat. No. 406/0179/00). Hematoxylin and eosin (H&E) was performed using the Vector Laboratories H&E Kit (Cat. No. 3502), as per their recommended protocol and mounted using Vectamount (Vector Laboratories, Cat. No. H5000-60).

Murine heads were decalcified and processed as described previously[51]. Following paraffin wax embedding, 2.5 µm sections were cut onto poly-L-silane-coated slides and stained with Gill's hematoxylin and Putt's eosin (both made in-house). Slides were imaged on a NanoZoomer Digital Pathology (NDP) slide scanner (Hamamatsu) and analyzed with NDP.view 2 software.

**Karyotype**. G-band analysis was performed on metaphase spreads obtained after 24 h unstimulated culture (RPMI-1640, Colcemid, 5% CO$_2$). Cells were harvested, slides made according to the laboratory standard-operating procedure. G-band staining has been done with the use of an automated staining machine (Leica Autostainer XL). Karyotype analysis was performed with the use of CytoVision v 7.7 software (Leica Ltd). For each case, ten metaphase spreads were analyzed unless an insufficient number of metaphase spreads were found. A constitutional Robertsonian der(14;21)(q10;q10) was present in donor 2-derived $^{CRISPR}$MLL-AF4 ALL and was confirmed to be present in the original unedited FL cells from this donor (Supplementary Table 2).

**FISH**. FISH was carried out on interphase nuclei with the use of Cytocell FISH probes: *MLL-AF4* translocation, Dual Fusion Probe (Cytocell), *MLL-ENL* translocation, Dual Fusion Probe (Cytocell), *MLL-AF10* translocation, Dual Fusion Probe (Cytocell), *MLL-AF9* translocation, Dual Fusion Probe (Cytocell), and *MLL-AF6* translocation, Dual Fusion Probe (Cytocell) (Supplementary Table 5). FISH setup and wash were performed following the manufacturer's (Cytocell Ltd) standard protocol. Olympus BX41 fluorescent microscope equipped with the filters for FITC, Cy3, TexasRed, Aqua, DAPI, and double filter set for FITC/TexasRed was used for analysis. For each case, 200 interphase nuclei were examined and patterns scored.

**In vitro drug sensitivity assays**. Cryopreserved blast cells harvested from the spleen of $^{CRISPR}$MLL-AF4+ mice, *MLL*-r, or *ETV6-RUNX1*+ PDX ALL models, as well as SEM and KOPN8 cell lines, were assayed in vitro for their response to prednisolone and L-asparaginase using the MTT assay (Roche, Cell Proliferation Kit I) as described previously[34,35]. Briefly, cells were resuspended at $2 \times 10^6$ cells/ml in RPMI supplemented with 15% heat-inactivated FBS, 100 U/ml penicillin, 100 µg/ml streptomycin, and 1% ITS Liquid Media Supplement (Sigma). In flat-bottom 96-well plates, 100 µl cell suspension ($0.2 \times 10^6$ cells) was treated with a range of concentrations of prednisolone (Sigma, final concentrations: 0.05–900 µg/ml) or L-asparaginase (Cambridge Biosciences, final concentrations: 0.003–10 IU/ml) based on previously published studies. For prednisolone, dimethyl sulfoxide was added to untreated control wells.

After 48 h incubation (or 96 h incubation for SEM and KOPN8 cell lines) at 37 °C and 5% CO$_2$, 10 µl MTT reagent was added to each well, after which the plates were incubated for another 4 h. During this time, the tetrazolium salt MTT is reduced into a purple-colored formazan product by living but not dead cells. One hundred microliters of Solubilization Solution was added to each well to dissolve the formazan crystals. The optical density (OD) of each well was measured on a SPECTROstar Nano (BMG Labtech) microplate reader at 570 nm. For each drug concentration, leukemia cell survival (LCS) was calculated by the following equation: LCS = (OD treated well/OD untreated well) × 100%. Drug resistance was expressed by the LC50, the drug concentration lethal to 50% of the cells.

**RT-qPCR**. Total RNA was extracted from cells using an RNeasy Micro Kit (Qiagen). Complementary DNA (cDNA) was generated from polyA messenger RNA (mRNA) using a SuperScript III Kit (Invitrogen). qPCR was carried out on cDNA using SYBRGreen Master Mix (Thermo Fisher) and a QuantStudio3 Real-Time PCR System (Thermo Fisher). For a list of qPCR primers used, see Supplementary Table 3.

**CRISPR-Cas9 off-target editing analysis**. Potential off-target editing sites for both MLL and AF4 sgRNAs were predicted using the Synthego CRISPR guide verification tool (https://design.synthego.com/#/validate). None of the potential off-target sites had >3 mismatches in the guide sequence. To test off-target sites with three mismatches (total of nine loci), genomic DNA was extracted from the spleen of primary recipient $^{CRISPR}MLL$-AF4+ mice ($n = 3$ mice from two individual FL samples) and from biologically matched, unedited FL cells ($n = 2$ individual FL samples) using a DNeasy Blood and Tissue Kit (Qiagen). An ~1 kb region of DNA around the target cut site was amplified by PCR and Sanger sequenced (Eurofins). Sanger-sequencing traces from primary $^{CRISPR}MLL$-AF4+ samples were compared to matched wt FL traces using the ICE Analysis online tool (Synthego, https://ice.synthego.com). The absence of off-target editing is reported as the percentage of reads in the $^{CRISPR}MLL$-AF4+ edited cells that match the unedited cells at each of these loci.

**RNA-sequencing**. Approximately $3 \times 10^5$ CD45+CD19+ cells were sorted from the BM of three primary $^{CRISPR}MLL$-AF4+ recipient mice and three control primary recipient mice (Cas9 control, $n = 2$; Cas9 plus MLL-sgRNA, $n = 1$). Total RNA was extracted using an RNeasy Mini Kit (Qiagen). Poly(A) purification was conducted using the NEB Poly(A) mRNA magnetic isolation module as per the manufacturer's protocol. Library preparation was carried out using the Ultra II Directional RNA Library Prep Kit (NEB, E7765). RNA libraries were sequenced by paired-end sequencing using a 150-cycle high output kit on a Nextseq 500 (Illumina). RNA-seq protocols for sorted subpopulations of FL HSPC have been described previously in ref. [27].

**IgH rearrangement analysis**. Samples were screened for IgH complete (VH–DH–JH) and IgH incomplete (DH–JH) rearrangements using BIOMED-2 protocols to detect clonality. DNA was extracted from cells from the BM of three primary $^{CRISPR}MLL$-AF4+ recipient mice. IgH rearrangements were analyzed as described in ref. [45].

**ChIP-sequencing**. The full protocol is described in ref. [38]. In short, up to $5 \times 10^7$ cells were sonicated (Covaris) following the manufacturer's protocol and incubated with an antibody overnight. Magnetic protein A and G beads (Thermo Fisher Scientific) were used to isolate antibody–chromatin complexes. Antibodies used are detailed in Supplementary Table 4. Beads were washed three times using a solution of 50 mM HEPES-KOH (pH7.6), 500 mM LiCl, 1 mM EDTA, 1% NP40, and 0.7% sodium deoxycholate and once with Tris-EDTA. Samples were eluted and proteinase K/RNase A-treated. Samples were purified using a ChIP Clean and Concentrator Kit (Zymo). DNA libraries were generated using the NEBnext Ultra DNA Library Preparation Kit for Illumina (NEB, E7103). Libraries were sequenced by paired-end sequencing using a 75-cycle high output kit on a Nextseq 500 (Illumina).

**NGS analysis**. For RNA-seq, following sequencing, QC analysis was conducted using the fastQC package. Reads were mapped to the human genome assembly using STAR (https://github.com/alexdobin/STAR/). The featureCounts function from the Subread package (http://subread.sourceforge.net/) was used to quantify gene expression levels using standard parameters. This was used to identify differential gene expression globally using the edgeR package (https://bioconductor.org/packages/release/bioc/html/edgeR.html). Differential gene expression was defined by an adjusted $p$ value (FDR) of <0.05. Infant-ALL RNA-seq datasets were analyzed as described previously[45]. GSEA analysis was performed using the fgsea function in the fgsea R package to determine the positive enrichment score and enrichment $p$ value of gene sets within differentially expressed genes, nperm = 1000[52].

To derive an FL vs ABM gene signature, bulk RNA-seq for sorted subpopulations of FL HSPC[27] were compared to matched sorted subpopulations of ABM HSPC[28] (FL HSC vs adult BM HSC, FL MPP vs ABM MPP, FL LMPP vs ABM LMPP, and FL CBPs vs ABM CLP). Genes that were differentially expressed between FL and ABM in at least one matched HSPC subpopulation were included in the gene signature. Genes that showed a significant change in opposite directions in different HSPC subtypes (e.g., upregulated in FL HSC vs ABM HSC, but downregulated in FL LMPP vs ABM LMPP) or in the normal vs leukemic setting (e.g., upregulated in FL HSPC vs ABM HSPC, but downregulated in MLL-AF4 infant-ALL vs MLL-AF4 childhood-ALL) were filtered out of the gene signature to leave a total of 5709 genes (Supplementary Data 2).

To analyze the effects of HOXA status and age on MLL-AF4 ALL blasts, we used the generalized linear model (glm) functionality of the edgeR package to carry out a three-way comparison between $HOXA^{lo}$ MLL-AF4 infant-ALL, $HOXA^{hi}$ MLL-AF4 infant-ALL, and $HOXA^{hi}$ MLL-AF4 childhood-ALL. This identified 765 marker genes, which could differentiate the three patient subsets from one another. UMAP analysis including the patient samples and $^{CRISPR}MLL$-AF4+ ALL was carried out based on this 765 gene signature.

For ChIP-seq, quality control of FASTQ reads, alignment, PCR duplicate filtering, blacklisted region filtering, and UCSC data hub generation was performed using an in-house pipeline as described[53]. The HOMER (http://homer.ucsd.edu/homer/) tool makeBigWig.pl command was used to generate bigwig files for visualization in UCSC, normalizing tag counts to tags per $1 \times 10^7$. ChIP-seq peaks were called using the

HOMER tool findPeaks.pl with a ChIP input sample used to estimate background signal. Gene profiles were generated using the HOMER tool annotatePeaks.pl.

**Statistics**. Two-tailed Mann–Whitney, log-rank (Mantel–Cox) tests, and analysis of variance followed by multiple comparisons testing were used to compare experimental groups as indicated in the figure legends. Statistical analyses were performed using GraphPad Prism v7.04 or R v4.0.1. Data are expressed as mean ± SEM, unless otherwise indicated.

**Reporting summary**. Further information on research design is available in the Nature Research Reporting Summary linked to this article.

## Data availability
The RNA-seq and ChIP-seq data generated in this study have been deposited in the NCBI GEO database under accession code NCBI GEO: GSE162041. These data are included in Figs. 1, 4, and 5 and Supplementary Figs. 1, 2, and 6. Source data are provided with this paper.

## Code availability
ChIP-seq data were analyzed using an in-house pipeline[53]. Further information and requests for resources and reagents may be directed to and will be fulfilled by the corresponding authors.

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

## Acknowledgements

T.A.M., S.R., N.T.C., and N.F. were funded by Medical Research Council (MRC, UK) Molecular Haematology Unit Grants MC_UU_12009/6 and MC_UU_00016/6. S.O'B. was funded by the Department of Paediatrics and Alexander Thatte Fund, University of Oxford. A.R. was supported by a Bloodwise Clinician Scientist Fellowship (grants: 14041 and 17001), Wellcome Trust Clinical Research Career Development Fellowship (216632/Z/19/Z), Lady Tata Memorial International Fellowship, and EHA-ASH Translational Research Training in Haematology Fellowship. I.R. is supported by the NIHR Oxford BRC, by a Bloodwise Program Grant (13001), and by the MRC Molecular Hematology Unit (MC_UU_12009/14). T.J. was supported as part of Wellcome Trust CRCDF (216632/Z/19/Z). C.H. was funded by the Little Princess Trust & Children's Cancer and Leukaemia group (CCLG 2017-13). N.E. was supported as part of Blood Cancer UK (1259; myeloid preleukaemia of Down syndrome). We gratefully acknowledge the Translational Histopathology Laboratory (THL) at the CRUK Oxford Centre, Department of Oncology for processing, sectioning, and staining spleen and liver tissue samples and Lynn Stevenson and Clare Orange, University of Glasgow, for brain histology and imaging. We would also like to acknowledge the WIMM Flow Cytometry Facility, which is supported by the MRC HIU; MRC MHU (MC_UU_12009); NIHR Oxford BRC; Kay Kendall Leukemia Fund (KKL1057), John Fell Fund (131/030 and 101/517), the EPA fund (CF182 and CF170) and by the WIMM Strategic Alliance awards G0902418 and MC_UU_12025. We thank the High-Throughput Genomics Group at the Wellcome Trust Centre for Human Genetics (funded by Wellcome Trust Grant Reference 090532/Z/09/Z); the MRC WIMM Centre for Computational Biology (CCB), Radcliffe Department of Medicine, University of Oxford; and Jelena Telenius for the use of her pipelines. We gratefully acknowledge helpful advice from Professor Anthony Moorman, Newcastle University during revisions of the manuscript. The human fetal material was provided by the Joint MRC/Wellcome Trust Grant 099175/Z/ 12/Z Human Developmental Biology Resource (http://hdbr.org). We thank Prof Pablo Menendez for helpful advice, and gratefully acknowledge the kind generosity of patients, their parents, and staff at Great Ormond Street Hospital, London.

## Author contributions

S.R., I.R., T.A.M., and A.R. conceived the experimental design; S.R., T.J., N.T.C., N.F., N.E., S.O'B., M.d.M.L.F., D.A., T.C., C.D, S.I., D.L., G.W., and C.H. carried out experiments; S.R., N.T.C., T.J., C.H., and A.R. analyzed and curated the data; S.R., C.H., I.R., T.A.M., and A.R. interpreted the data; S.R., I.R., T.A.M., and A.R. wrote the original manuscript; S.R., N.T.C., T.J., S.I., J.B., P.A., A.J.M., C.H., I.R., T.A.M., and A.R. contributed to reviewing and editing the manuscript. I.R., A.R., and T.A.M. provided supervision and funding.

## Competing interests

T.A.M. is a founding shareholder of OxStem Oncology (OSO), a subsidiary company of OxStem Ltd. T.A.M. and N.T.C. are founding shareholders and paid consultants for Sandymount Therapeutics (a subsidiary company of Dark Blue Therapeutics). S.O'B. is now an employee of Becton, Dickinson and Company (BD). S.O'B.'s contributions to the work were made prior to the commencement of employment at BD. The other authors declare no competing interests.
