## [Peer Review File · Nature Communications]

A human fetal liver-derived infant MLL-AF4 Acute Lymphoblastic Leukemia model reveals a distinct fetal gene expression programREVIEWER COMMENTS

Reviewer #1 (Remarks to the Author):

The manuscript by Rice and co-workers for the first time comprehensively demonstrates the importance and influence of the fetal liver HSC as the cell of origin of MLL-AF4+ pro-B ALL in infants, and with that for the first time confirm a hypothesis shared by many in the field. On top of that, Rice et al have generated the first fully humanised mouse model of MLL-AF4+ infant ALL leukemogenesis, that accurately recapitulates the human disease, and even leads to leukaemia CNS infiltration, which is rather characteristic for this type of leukaemia. The manuscript is well written, and the experiments and statistical analyses all are sound and appropriate. This manuscript represents an important contribution to our understanding of how this aggressive type of leukaemia develops. I'd like to congratulate the authors with this excellent study. I do, however, have a few modest questions/suggestions:

1. If I understood correctly, the CRISPR MLL-AF4 cells, as well as the leukaemia cells coming forth from the mouse model, expressed both MLL-AF4 and the reciprocal AF4-MLL. Did the authors also attempt to induce MLL-AF4 driven leukemogenesis in the absence of AF4-MLL, or even with AF4-MLL expression alone (in the absence of MLL-AF4)? Some insights into this matter would certainly help, as there has been some debate regarding the role of the reciprocal AF4-MLL.
2. It is very clear that the presented mouse model nicely recapitulates infant MLL-AF4+ pro-B ALL from the (immuno)phenotypic characteristics, the manner in which the leukaemia aggressively develops in mice, and the gene expression analysis. It would, however, be a nice addition if the authors could show that the leukaemia cells coming forth from their model also resembles the typical refractory drug response profile of MLL-AF4+ pro-B infant ALL to chemotherapeutic drugs currently used in the treatment of this malignancy: i.e. are these cells, like the cells from actual patients, particularly resistant to prednisone/prednisolone and L-asparaginase?

Reviewer #2 (Remarks to the Author):

Starting with the well-known concept that the aggressive nature and genetic simplicity of infant MLL-r ALL is largely due to its prenatal origin, Rice et al. analyze previously published datasets (transcriptomics from infant vs childhood ALL and fetal vs adult HSPCs) in order to identify a fetal expression signature in infant ALL that demonstrates its prenatal origin and suggests that the expression of some of those genes is maintained in blasts.

To functionally demonstrate that MLL-AF4+ infant ALL can indeed initiate from a fetal context, the authors then use an elegant CRISPR-Cas9 system to induce the t(4;11) translocation in human fetal cells, which upon transplantation cause an aggressive B-ALL with phenotypic and molecular features of infant ALL. This novel model is really the star of this paper, which should be reflected in the way this manuscript is presented, including the title, as it shows that indeed the t(4;11) translocation can initiate aggressive B-ALL on its own and provides the community with an important tool, while some of the molecular analyses distract from this, especially since some of it has already been published and the conclusions are not always justified.

Specific comments:

1. The authors analyze the dataset from Andersson et al (ref. 15) to show that there are two subgroups of infant MLL-AF4 ALL, characterized by the opposite expression of HOXA and IRX genes, one subgroup of which appears to be more closely related to childhood ALL. This exact analysis and findings have recently been published by Symeonidou et al. (doi: 10.1016/j.exphem.2020.10.002). In fact, Fig. 1a and Supp Fig.1a are identical to Fig. 1a in the Symeonidou paper and should really be removed, and the authors must reference this paper (including in line 32 of the introduction).
2. Considering that the authors noted these different subgroups of infant ALL, it is surprising that they did not incorporate this into their subsequent analysis when they are attempting to identify an infant-

specific signature. It appears that they pooled all infant cases for their comparison to the childhood cases, even though the IRXlo/HOXAhi subset clearly clusters with the childhood cases. Many genes, the expression of which might be crucial to the 'true' infant subset, which is also known to have a worse prognosis, may have been missed that way. Pooling the IRXlo/HOXAhi subset with the childhood cases or just comparing the two infant subsets (as done in the Symeonidou et al paper), may deliver a more representative infant-specific signature.

3. In line 103 on page 6, the authors suggest that MLL-AF4-driven B lineage specification occurs at a progenitor stage, based on a higher co-expression of CD34 and CD19; however, the number of CD34+ cells in the CRISPR-MLL-AF4 model seems to be extremely low (Fig. 2c,d). What was the absolute number of Lin-CD34+ cells and does it allow such a statement to be made?

4. While the CNS infiltration gives the authors' model immense credit, the sentence on p. 7, line 126/127 "...a key clinical feature of infant-ALL that has not been previously reported in MLL-AF4 mouse models." should be removed. It is misleading as it implies that none of the other models have had CNS infiltration, when it is more likely that it had been missed as its detection requires an expert like Dr Halsey. It is very likely that the model from Michael Thirman's group (ref. 27) with its strong pro-B ALL phenotype (CD19+ CD10-) would also have shown CNS infiltration as it is quite common with human ALL cells. A similar statement should also be removed from the Discussion.

5. The variability in the CD34 phenotype in the in vivo model was interesting, with a lot more CD34+ cells in the proB-ALL, but still less than in patients. What was it like in the preB-ALL? Does CD34 positivity increase in serial transplants (the CD34 phenotype was not included in Supp. Table 3)? Perhaps the authors could add a statement about the likely significance of the CD34 expression.

6. There is a mistake in the legends to Supp. Fig. 4g. The flow plots are only from one mouse, but the figure legend states control and CRISPR-MLL-AF4. I assume it is just the control?

7. Fig. 4a: it is not surprising that the CRISPR-MLL-AF4 samples cluster with the other MLL-AF4 samples (rather than MLLwt) as they carry the same translocation, while MLLwt disease is molecularly quite distinct. Maybe MLLwt was not the best example/control to choose. It would be very useful though, if the authors could colour the IRXlo/HOXAhi and the IRXhi/HOXAlo MLL-AF4 iALL subsets in different colours, especially since one of the CRISPR-MLL-AF4 samples is quite different.

8. The fact that the 3 CRISPR-MLL-AF4 samples are indeed quite different is somewhat glossed over, but may be potentially interesting. Is it to do with the preB vs proB phenotype? Is the preB sample the distinct one in Fig. 4a and 5a and the one with the higher HOXA9 expression and absent IRX1 expression in Supp. Fig. 5c? Because of the heterogeneity of the expression in Supp. Fig. 5c, the statement on p.8. line 152 "Moreover, CRISPR-MLL-AF4+ ALL resembled HOXAlo/IRXhi MLL-AF4 infant-ALL" is not true.

9. I am not convinced that the authors' hypothesis that the CB-derived MLL-Af4 samples represent chALL is correct. In Fig. 5a, they cluster as closely to chALL as they cluster to iALL – in fact, one of the CRISPR-MLL-AF4 samples is as close to the chALL cluster as the CB MLL-Af4 samples.

10. While there is a lot of evidence that the fetal cell of origin is an important contributing factor to the iALL phenotype, I don't think this was convincingly shown in this study. The fact that the CRISPR-MLL-AF4 samples clustered with the MLL-AF4 iALL samples and away from the CB MLL-Af4 samples is not surprising. It could easily be explained by the fact that the authors managed to induce a true t(4;11) translocation, including expression of the reciprocal AF4-MLL reciprocal fusion, that resembles the human disease situation much more closely than viral overexpression of a human:mouse MLL-Af4 chimaeric fusion. To demonstrate that it is entirely due to the fetal context, they would have to demonstrate that CRISPR-induced t(4;11) in CB cells produces a disease that clusters with the chALL samples. The fact that there are subtypes even within the infant MLL-AF4 ALL patients, some of which resemble childhood cases, suggests that it may be more complicated than that, and that the specific progenitor subtype may be equally important. This may also explain why the authors obtained both proB as well as preB phenotypes from fetal cells – they may have induced the translocation in different progenitor types within the broad CD34+ population. This should be acknowledged and discussed.

We would like to thank the reviewers for their positive and constructive comments on our paper. In response to these, we have generated some new data and rewritten parts of the text and we feel that the paper has significantly improved as a result. We have provided a specific point-by-point response below and marked all changes in the text in blue font in the manuscript.

Summary of changes to figures and tables:

Figure 1: Original Figure 1a removed as suggested by reviewer 2

- (a) Previously Fig 1b. New annotation on heatmap
- (b) Previously Fig 1c
- (c) Previously Fig 1d. New panel added
- (d-e) Previously Figure 1e-f

Figure 2: (b) *AF4-MLL* RT-qPCR now n=3

- (c) Font size increase

Figure 3: (e) Font size increase

Figure 4: (a) *MLL-AF4* ALL patients recolored by *HOXA* status. ^{CRISPR}*MLL-AF4+* ALL datapoints: borders recolored by immunophenotype

- (c) New panel
- (d) Previously Supplementary Figure 4c (plotted as TPM rather than log2-transformed TPM)
- (e-f) Previously Fig 4c-d

Figure 5: (a) ^{CRISPR}*MLL-AF4+* ALL datapoints: borders recolored by immunophenotype

- (d) ^{CRISPR}*MLL-AF4+* ALL datapoints: borders recolored by immunophenotype

Supplementary Figure 1: original Supplementary Figure 1b removed

Supplementary Figure 1: (a-b) Previously Supplementary Figure 1c-d

Supplementary Figure 1: (c) Previously Supplementary Figure 1a

Supplementary Figure 2: (c) FDR added to plots

Supplementary Figure 3: (c) New panel

Supplementary Figure 4: (f) New panel

Supplementary Figure 5: (a-c) Previously Supplementary Figure 4f-h. New data included in Supplementary Figure 5a.

Supplementary Figure 5: (d) New panel

Supplementary Figure 6 (previously Supplementary Figure 5):

- (b) *MLL-AF4* ALL patient recolored by *HOXA* status
- (c) Previously Supplementary Figure 4d.

Supplementary Table 4: New table

Supplementary Table 5-6: Previously Supplementary Table 4-5

Reviewers' comments:

Reviewer #1 (Remarks to the Author):

The manuscript by Rice and co-workers for the first time comprehensively demonstrates the importance and influence of the fetal liver HSC as the cell of origin of *MLL-AF4*+ pro-B ALL in infants, and with that for the first time confirm a hypothesis shared by many in the field. On top of that, Rice et al have generated the first fully humanised mouse model of *MLL-AF4*+ infant ALL leukaemogenesis, that accurately recapitulates the human disease, and even leads to leukaemia CNS infiltration, which is rather characteristic for this type of leukaemia. The manuscript is well written, and the experiments and statistical analyses all are sound and appropriate. This manuscript represents an important contribution to our understanding of how this aggressive type of leukaemia develops. I'd like to congratulate the authors with this excellent study. I do, however, have a few modest questions/suggestions:

1. If I understood correctly, the ^{CRISPR}*MLL-AF4*+ cells, as well as the leukaemia cells coming forth from the mouse model, expressed both *MLL-AF4* and the reciprocal *AF4-MLL*. Did the authors also attempted to induced *MLL-AF4* driven leukaemogenesis in the absence of *AF4-MLL*, or even with *AF4-MLL* expression alone (in the absence of *MLL-AF4*)? Some insights into this matter would certainly help, as there has been some debate regarding the role of the reciprocal *AF4-MLL*.

Thank you very much for your positive comments on our paper. We agree with the reviewer that the contribution of the reciprocal *AF4-MLL* fusion gene to leukemogenesis is indeed a very interesting question, which is still a topic of debate in the field. However, our current modelling approach does not appear to be conducive to creating single fusion genes. All of the ^{CRISPR}*MLL-AF4*+ translocation events we have analyzed here (n=6, combined *in vitro* and *in vivo* experiments) expressed both *MLL-AF4* and *AF4-MLL*. In order to address the role of individual fusion genes in leukemia initiation, it would be necessary to design a method to simultaneously induce the translocation and introduce a premature STOP codon, or potentially delete either *MLL-AF4* or *AF4-MLL*, perhaps through homology-directed repair (HDR). While this is something we would be interested in pursuing in the future, we feel it is beyond the scope of this paper.

We have included the following discussion of this point in lines 265-270, which we hope will provide greater clarity regarding reciprocal fusion gene expression in ^{CRISPR}MLL-AF4+ ALL:

“Finally, all ^{CRISPR}MLL-AF4+ ALLs showed expression of both reciprocal fusion genes, MLL-AF4 and AF4-MLL. The contribution of AF4-MLL to the initiation of MLL-AF4 leukemia has been a topic of debate in the field. However, our editing approach has not allowed us to address the question of the relative importance of AF4-MLL to transformation. In the future, our editing approach could potentially be adapted to express only one or both of the reciprocal fusion genes.”

2. It is very clear that the presented mouse model nicely recapitulates infant MLL-AF4+ pro-B ALL from the (immuno)phenotypic characteristics, the manner in which the leukaemia aggressively develops in mice, and the gene expression analysis. It would, however, be a nice addition if the authors could show that the leukaemia cells coming forth from their model also resembles the typical refractory drug response profile of MLL-AF4+ pro-B infant ALL to chemotherapeutic drugs currently used in the treatment of this malignancy: i.e. are these cells, like the cells from actual patients, particularly resistant to prednisone/prednisolone and L-asparaginase?

Chemo-resistance is a key feature of MLL-r ALL and we thank the reviewer for this interesting suggestion.

We have now added new data (Supplementary Figure 4f) in which drug sensitivity of ALL blasts from patient-derived xenograft (PDX) models, as well as the MLL-r SEM and KOPN8 cell lines, was compared with ^{CRISPR}MLL-AF4+ ALL cells. PDX blasts, cell lines and ^{CRISPR}MLL-AF4+ blasts were cultured in a range of prednisolone and L-asparaginase concentrations for 48 hours (primary) to 96 hours (cell lines), followed by cytotoxicity analysis using the MTT assay. We found that the ^{CRISPR}MLL-AF4+ ALL cells were indeed amongst the most prednisolone-resistant samples we analyzed, and LC50 values were consistent with previously reported *in vitro* responses for chemo-resistant patient samples¹⁻³. We used PDX models as comparators because we believed a primary leukemia passaged through an NSG mouse provided the most appropriate comparison to our ^{CRISPR}MLL-AF4+ ALL that was derived after xenograft. The PDX models included 4 ETV6-RUNX1 ALLs and 2 MLL-r ALLs. Although all of the samples were derived from children >1 year of age, it has been previously demonstrated that MLL-r and/or proB ALLs at all ages can be prednisolone resistant³.

These new data have now been included in Supplementary Figure 4f and Supplementary Table 3, and discussed in lines 139-144:

“As chemo-resistance is also an important feature of *MLL-r* infant-ALL, we compared the responses of *CRISPR* *MLL-AF4+* ALL blasts to prednisolone and L-asparaginase with ALL patient-derived xenograft (PDX) samples and the SEM and KOPN8 cell lines, and found similar levels of *in vitro* drug-resistance to previous reports of treatment-resistant patient samples²⁹⁻³¹ (Supplementary Figure 4f, Supplementary Table 3).”

Reviewer #2 (Remarks to the Author):

Starting with the well-known concept that the aggressive nature and genetic simplicity of infant *MLL-r* ALL is largely due to its prenatal origin, Rice et al. analyze previously published datasets (transcriptomics from infant vs childhood ALL and fetal vs adult HSPCs) in order to identify a fetal expression signature in infant ALL that demonstrates its prenatal origin and suggests that the expression of some of those genes is maintained in blasts.

To functionally demonstrate that *MLL-AF4+* infant ALL can indeed initiate from a fetal context, the authors then use an elegant CRISPR-Cas9 system to induce the t(4;11) translocation in human fetal cells, which upon transplantation cause an aggressive B-ALL with phenotypic and molecular features of infant ALL. This novel model is really the star of this paper, which should be reflected in the way this manuscript is presented, including the title, as it shows that indeed the t(4;11) translocation can initiate aggressive B-ALL on its

“A novel human fetal liver-derived model reveals that MLL-AF4 drives a distinct fetal gene expression program in infant-ALL”

own and provides the community with an important tool, while some of the molecular analyses distract from this, especially since some of it has already been published and the conclusions are not always justified.

In response to the reviewer’s suggestion that the model itself should be highlighted more in the title, we have revised the title to:

“A novel human fetal liver-derived model reveals that MLL-AF4 drives a distinct fetal gene expression program in infant-ALL”

Specific comments:

1. The authors analyze the dataset from Andersson et al (ref. 15) to show that there are two subgroups of infant *MLL-AF4* ALL, characterized by the opposite expression of *HOXA* and *IRX* genes, one subgroup of which appears to be more closely related to childhood-ALL. This exact analysis and findings have recently been published by Symeonidou et al. (doi: 10.1016/j.exphem.2020.10.002). In fact, Fig. 1a and Supp Fig.1a are identical to Fig. 1a in the Symeonidou paper and should really be removed, and the authors must reference this paper (including in line 32 of the introduction).

We thank the reviewer for highlighting these similarities. We agree that the UMAPs in Figure 1a and Supplementary Figure 1b (we think the reviewer is referring to this rather than Supplementary Figure 1a) are very similar to the PCA produced in Figure 1a of Symeonidou et al. and therefore are unnecessary to include here. We have removed these UMAPs and cited Symeonidou et al. in the appropriate locations, including in line 29 in Introduction. Instead, we have added an annotation to the heatmap in new Figure 1a (previously Figure 1b) to identify *HOXA*^{lo}/*HOXA*^{hi} patients. We included this annotation to highlight that the infant-ALL signature separates infant-ALL from childhood-ALL regardless of other well-known molecular characteristics, such as *HOXA* status. Please see comment 2 below for further discussion.

“A novel human fetal liver-derived model reveals that *MLL-AF4* drives a distinct fetal gene expression program in infant-ALL”

2. Considering that the authors noted these different subgroups of infant ALL, it is surprising that they did not incorporate this into their subsequent analysis when they are attempting to identify an infant-specific signature. It appears that they pooled all infant cases for their comparison to the childhood cases, even though the *IRX⁰/HOXA^{hi}* subset clearly clusters with the childhood cases. Many genes, the expression of which might be crucial to the ‘true’ infant subset, which is also known to have a worse prognosis, may have been missed that way. Pooling the *IRX⁰/HOXA^{hi}* subset with the childhood cases or just comparing the two infant subsets (as done in the Symeonidou et al paper), may deliver a more representative infant-specific signature.

We thank the reviewer for making these important points. We agree with the reviewer that infant-ALL can be split into 2 distinct subsets based on *HOXA/IRX* gene expression profiles. However, we feel that there is not enough evidence to suggest that *HOXA^{lo}* infant-ALL is the “true” infant-ALL signature, as both of these infant-ALL subsets appear to be distinct from childhood-ALL (please see our revised Fig 1a) based on 617 genes differentially expressed in *MLL-AF4* infant-ALL vs *MLL-AF4* childhood-ALL. We think this suggests that *HOXA* status *per se* is unlikely to drive age-related differences, although it is an important way to identify distinct functional subsets within the infant-ALL group itself. Our intention when analyzing the patient dataset was to try and define a gene signature that separates infant-ALL from childhood-ALL irrespective of their *HOXA* status and to identify the gene expression program that does drive these age-related differences. As such, this is why we compared all *MLL-AF4* infant-ALL patients to *MLL-AF4* childhood ALL.

We have now revised the text in this section of the results (lines 49-67) to more clearly explain the goal of our analysis and our intention in annotating the *HOXA* status of the patients. We have also revised Figure 1c to demonstrate that there are no significant differences between *HOXA^{lo}/HOXA^{hi}* *MLL-AF4* infant-ALL in the expression of the top 10 most significantly upregulated genes in the infant-ALL signature, even though this signature is clearly able to distinguish infant-ALL from childhood-ALL.

To address this point in more detail, we performed a 3-way statistical comparison between *HOXA*^{lo} *MLL-AF4* infant-ALL, *HOXA*^{hi} *MLL-AF4* infant-ALL and *HOXA*^{hi} *MLL-AF4* childhood-ALL and this has been added as Supplementary Table 4 and a new Figure 4c. We found both classic *HOXA/IRX* signature genes and age-related signature genes included in the differentially expressed gene list; and UMAP analysis based on these 1,427 genes showed a continuum from *HOXA*^{hi} *MLL-AF4* childhood-ALL to *HOXA*^{lo} *MLL-AF4* infant-ALL (Figure 4c). Of note, a technical limitation with such analyses is that, as no *HOXA*^{lo} childhood-ALL patients were present in this dataset, the *HOXA/IRX* signature masks the age-related signature using this approach. However, we did find this analysis useful to address concerns raised in regards to the *HOXA* status of all *CRISPR*^{MLL-AF4+} ALLs. Please see comments 7 and 8 below for further discussion.

3. In line 103 on page 6, the authors suggest that *MLL-AF4*-driven B lineage specification occurs at a progenitor stage, based on a higher co-expression of CD34 and CD19; however, the number of CD34+ cells in the *CRISPR*^{MLL-AF4+} model seems to be extremely low (Fig. 2c,d). What was the absolute number of Lin-CD34+ cells and does it allow such a statement to be made?

We thank the reviewer for raising this point, and we have now added data for the absolute number of Lin-CD34+ cells throughout culture as Supplementary Figure 3c. There were similar numbers of CD34+ cells from week 3 in *CRISPR*^{MLL-AF4+} and control cultures. We think that *MLL-AF4*-driven B-lineage specification might occur at a progenitor stage because there is still a striking increase in the proportion of CD19+ B progenitors in *CRISPR*^{MLL-AF4+} cultures at these timepoints. We have changed the wording in the text to reflect the fact that we have not addressed this directly (lines 112-116).

“A novel human fetal liver-derived model reveals that MLL-AF4 drives a distinct fetal gene expression program in infant-ALL”

“Although the proportion of CD34+ cells in $CRISPR^{MLL-AF4+}$ cultures was reduced, the absolute number of CD34+ cells was comparable between $CRISPR^{MLL-AF4+}$ and control cultures from week 3 (Supplementary Figure 3c). The majority of $CRISPR^{MLL-AF4+}$ CD34+ cells were CD19+ B progenitors, suggesting that MLL-AF4-driven B lineage specification might occur at a progenitor stage (Figure 2d right).”

4. While the CNS infiltration gives the authors' model immense credit, the sentence on p. 7, line 126/127 "...a key clinical feature of infant-ALL that has not been previously reported in MLL-AF4 mouse models." should be removed. It is misleading as it implies that none of the other models have had CNS infiltration, when it is more likely that it had been missed as its detection requires an expert like Dr Halsey. It is very likely that the model from Michael Thirman's group (ref. 27) with its strong pro-B ALL phenotype (CD19+ CD10-) would also have shown CNS infiltration as it is quite common with human ALL cells. A similar statement should also be removed from the Discussion.

We agree with the reviewer and have now rephrased our statements on lines 137-139:

“ $CRISPR^{MLL-AF4+}$ mice also had central nervous system (CNS) disease, with extensive parameningeal blast cell infiltration (Figure 3d); a key clinical feature of infant-ALL.”

and lines 271-279:

“As well as providing insights into MLL-AF4 function in a human fetal cell context, $CRISPR^{MLL-AF4+}$ ALL provides a previously lacking, preclinical model for translational studies that specifically recapitulates poor prognosis infant-ALL. For example, the CNS disease observed in $CRISPR^{MLL-AF4+}$ ALL is a common clinical feature of infant-ALL that can lead to CNS relapse in these patients. Therefore, the ability of novel treatments to eradicate blasts from the CNS is an important consideration, and this can be tested in $CRISPR^{MLL-AF4+}$ ALL. “

5. The variability in the CD34 phenotype in the *in vivo* model was interesting, with a lot more CD34+ cells in the proB-ALL, but still less than in patients. What was it like in the preB-ALL? Does CD34 positivity increase in serial transplants (the CD34 phenotype was not included in Supp. Table 3)? Perhaps the authors could add a statement about the likely significance of the CD34 expression.

We thank the reviewer for asking for clarification, as we did not include sufficient data in our first submission to represent the full pattern of CD34 expression in our model and in patient samples. To address this, we have included several pieces of additional data. Firstly, we

Rice et al Reviewer Response

“A novel human fetal liver-derived model reveals that MLL-AF4 drives a distinct fetal gene expression program in infant-ALL”

now show flow plots for all 3 primary ^{CRISPR}MLL-AF4+ mouse BM. This shows that CD34 expression is unrelated to CD10 expression in our model, as 1/2 proB ALLs and the preB ALL were CD34- (Supplementary Figure 5a). This is in keeping with data from primary MLL-r patients, where CD34 surface expression is known to be heterogeneous, with some infant-ALL patients having CD34- blasts. Secondly, we show summary data of CD34 expression in the BM of MLL-r infant-ALL and childhood-ALL patients, as well as primary and secondary ^{CRISPR}MLL-AF4+ mouse BM (new Supplementary Figure 5d). This further shows that CD34 surface expression is heterogeneous among patients, regardless of age, and on average shows no significant difference. Our model recapitulates this.

The CD34 expression data from primary and secondary transplants is now included in Supplementary Table 3 as requested, and discussed in lines 157-161:

“The proportion of blasts that were CD34+ did not correlate with CD10 expression (Supplementary Figure 5a, Supplementary Table 3), nor did it increase significantly in secondary and tertiary recipients (Supplementary Figure 5d, Supplementary Table 3). This is in keeping with data from primary MLL-r patient samples, where CD34 expression is known to be heterogeneous (Supplementary Figure 5d).”

Unfortunately, due to COVID-related building closures, we were unable to harvest BM from tertiary transplants when they became moribund and were culled. However, we have included PB data for primary, secondary and tertiary recipients below (Reviewer response Figure 1) to demonstrate CD34 was heterogeneous even between secondary transplants derived from the same primary transplant, and that there was no significant difference in the proportion of CD34 blasts between all serial transplants.

	Primary		Secondary		Tertiary	
	Control	^{CRISPR} MLL-AF4+	Control	^{CRISPR} MLL-AF4+	Control	^{CRISPR} MLL-AF4+
Engrafted	5/5	3/3	1/1	4/4	2/2	3/3
ALL	0/5	3/3	0/1	4/4	0/2	3/3
Median latency	n/a	18 weeks	n/a	11.5 weeks	n/a	8 weeks
BM engraftment	5/5	3/3	1/1	4/4	2/2	3/3
Splenomegaly	0/5	3/3	0/1	4/4	0/2	3/3
CNS infiltration (/of those tested)	0/1	1/1	0/1	3/3	ND	ND
Pred resistant (LC50 > 100µg/ml)	n/a	1/1	n/a	2/2	ND	ND
L-asp resistant (LC50 > 0.1 IU/ml)	n/a	1/1	n/a	3/3	ND	ND
proB (CD19+CD10-CD20-IgM/IgD-)	n/a	2/3	n/a	2/4	n/a	3/3
preB (CD19+CD10+CD20-IgM/IgD-)	n/a	1/3	n/a	2/4	n/a	0/3
CD34 positive (>20% of blasts)	n/a	ProB (1/2); PreB (0/1)	n/a	ProB (1/2); PreB (2/2)	n/a	ND
MLL-AF4 expression	0/2	2/2	ND	ND	ND	ND
AF4-MLL expression	0/2	2/2	ND	ND	ND	ND
VDJ rearrangement	ND	2/3 (ProB and PreB) clonal; 1/3 (ProB) non-rearranged	ND	ND	ND	ND

a

Supplementary Figure 5a. Representative flow cytometry plots of viable, single cells in proB^{CRISPR} MLL-AF4⁺ (top) and preB^{CRISPR} MLL-AF4⁺ (bottom) BM at termination. (mCD45.1 = mouse CD45; hCD45 = human CD45).

d

Supplementary Figure 5d. Barplot showing the proportion of CD19⁺ B-ALL blasts that are CD34⁺ for primary MLL-r infant-ALL (dark green, n=8) and MLL-r childhood-ALL (orange, n=7) patient samples, and primary (n=3) and secondary (n=4) ^{CRISPR} MLL-AF4⁺ ALL (light green). Data shown as mean ± SEM.

CD34⁺ cells in ^{CRISPR} MLL-AF4⁺ PB

Reviewer response Figure 1. Proportion of CD19⁺ blasts that are CD34⁺ in ^{CRISPR} MLL-AF4⁺ mice through serial transplants. Data represent peripheral blood blasts from primary, secondary and tertiary mice. Each point is a mouse. Lines = secondary and tertiary mice are paired with the primary mouse from which they were derived to show variation through transplants. Data are shown as mean ± SEM.

“A novel human fetal liver-derived model reveals that *MLL-AF4* drives a distinct fetal gene expression program in infant-ALL”

6. There is a mistake in the legends to Supp. Fig. 4g. The flow plots are only from one mouse, but the figure legend states control and $^{CRISPR}MLL-AF4+$. I assume it is just the control?

Thank you for making this point. Supplementary Figure 4f and 4g are now part of a new Supplementary Figure 5 which includes the flow plots for all primary $^{CRISPR}MLL-AF4+$ ALLs (a), as well as extended flow data for $^{CRISPR}MLL-AF4+$ preB ALL (b), labelled “ $^{CRISPR}MLL-AF4+$ preB ALL (alternative panel)” in Supplementary Figure 5b.

7. Fig. 4a: it is not surprising that the $^{CRISPR}MLL-AF4+$ samples cluster with the other *MLL-AF4* samples (rather than *MLLwt*) as they carry the same translocation, while *MLLwt* disease is molecularly quite distinct. Maybe *MLLwt* was not the best example/control to choose. It would be very useful though, if the authors could colour the $IRX^{lo}/HOXA^{hi}$ and the $IRX^{hi}/HOXA^{lo}$ *MLL-AF4* iALL subsets in different colours, especially since one of the $^{CRISPR}MLL-AF4+$ samples is quite different.

Again, thank you for making this point. Please see our comments in response to point 2 above. In addition, we apologize for not making it clear that our intention for Figure 4a was mainly to show that $^{CRISPR}MLL-AF4+$ ALL was an accurate model of *MLL-r* (specifically *MLL-AF4*) ALL on the transcriptomic and epigenetic level. The comparison of $^{CRISPR}MLL-AF4+$ ALL with both *MLL-r* and *MLLwt* was our starting point and we were reassured that the $^{CRISPR}MLL-AF4+$ ALL clustered with the *MLL-r* samples, although we agree that this is not surprising. However, as well as recoloring the $HOXA^{hi}$ and the $HOXA^{lo}$ *MLL-AF4* infant-ALL subsets as the reviewer suggests (please see new Figure 4a and Supplementary Figure 6b), we have also performed some additional analyses.

Using a signature of 1,427 genes derived from a 3-way statistical analysis of $HOXA^{lo}$ *MLL-AF4* infant-ALL, $HOXA^{hi}$ *MLL-AF4* infant-ALL and $HOXA^{hi}$ *MLL-AF4* childhood-ALL (Supplementary Table 4, new Figure 4c), we performed clustering analysis of these patient samples as well as $^{CRISPR}MLL-AF4+$ ALL. We find that all $^{CRISPR}MLL-AF4+$ ALLs (proB and preB) represent the $HOXA^{lo}$ subset. As well as providing greater clarity as to the molecular characteristics of $^{CRISPR}MLL-AF4+$ ALL, it is perhaps surprising that both proB and preB ALL models are $HOXA^{lo}$. Moreover, it is interesting then that the *MLL-AF4* binding profile of $HOXA^{lo}$ $^{CRISPR}MLL-AF4+$ ALL is still so similar to the $HOXA^{hi}$ *MLL-AF4* SEM cell line.

Note: for clarity we have now used different borders to distinguish the proB and preB immunophenotypes for the $^{CRISPR}MLL-AF4+$ ALL datapoints in all figures (see details in response to comment 8).

Figure 4a. UMAP showing clustering of $^{CRISPR}MLL-AF4+$ (light green; black border = proB ALL, no border = preB ALL) and control (grey) mice with $HOXA^{lo} MLL-AF4$ (dark green), $HOXA^{hi} MLL-AF4$ (purple) and $MLLwt$ (blue) infant-ALL patient samples from a publicly available dataset²⁴ based on 7,041 significantly differentially expressed genes (FDR<0.05) between $^{CRISPR}MLL-AF4+$ ALL, controls, $MLL-AF4$ infant-ALL and $MLLwt$ infant-ALL.

Figure 4c. UMAP showing clustering of $^{CRISPR}MLL-AF4+$ (light green; black border = proB ALL, no border = preB ALL) with $HOXA^{lo} MLL-AF4$ infant-ALL, $HOXA^{hi} MLL-AF4$ infant-ALL and $HOXA^{hi} MLL-AF4$ childhood ALL from a publicly available patient dataset²⁰ based on 1,427 significantly differentially expressed genes (FDR<0.05) between these 3 patient subsets (Supplementary Table 4).

8. The fact that the 3 $^{CRISPR}MLL-AF4+$ samples are indeed quite different is somewhat glossed over, but may be potentially interesting. Is it to do with the preB vs proB phenotype? Is the preB sample the distinct one in Fig. 4a and 5a and the one with the higher $HOXA9$ expression and absent $IRX1$ expression in Supp. Fig. 5c? Because of the heterogeneity of the expression in Supp. Fig. 5c, the statement on p.8. line 152 “Moreover, $^{CRISPR}MLL-AF4+$ ALL resembled $HOXA^{lo}/IRX^{hi} MLL-AF4$ infant-ALL” is not true.

“A novel human fetal liver-derived model reveals that *MLL-AF4* drives a distinct fetal gene expression program in infant-ALL”

We agree with the reviewer that these are interesting findings. To better show how our results map to the proB and preB immunophenotypes of *CRISPR*^{MLL-AF4}, we have now modified the key to distinguish the preB from the proB ALLs in all molecular analyses and figures throughout the paper. In all UMAPs and barplots displaying gene expression data, proB *CRISPR*^{MLL-AF4} ALLs have a black border and preB *CRISPR*^{MLL-AF4} ALL has no border; we have also amended the figure legends accordingly. We apologize for the fact that our statement that *CRISPR*^{MLL-AF4} ALL resembled *HOXA*^{lo} *MLL-AF4* infant-ALL was misleading. In our original submission, we displayed *HOXA9* and *IRX1* expression as log2 transformed transcripts per million (TPM), which in hindsight made interpretation more difficult. In Figure 4d, we now show non-transformed TPM values. We hope it is now clearer that all *CRISPR*^{MLL-AF4} ALLs are *HOXA*^{lo}. The preB *CRISPR*^{MLL-AF4} ALL also shows high *IRX1* expression often associated with the *HOXA*^{lo} molecular profile. Moreover, we hope our clustering analysis in Figure 4c now makes it clearer that, based on a *HOXA*^{lo}/*HOXA*^{hi} signature of 1,427 genes, all *CRISPR*^{MLL-AF4} ALLs cluster with *HOXA*^{lo} *MLL-AF4* infant-ALL.

Figure 4d. Expression (TPM) of *HOXA9* and *IRX1* in *HOXA*^{hi} *MLL-AF4* infant-ALL (purple), *HOXA*^{lo} *MLL-AF4* infant-ALL (dark green) and *CRISPR*^{MLL-AF4+} ALL (light green; black border = proB ALL, no border = preB ALL). Data shown as mean ± SEM.

9. I am not convinced that the authors’ hypothesis that the CB-derived *MLL-Af4* samples represent chALL is correct. In Fig. 5a, they cluster as closely to chALL as they cluster to iALL – in fact, one of the *CRISPR*^{MLL-AF4+} samples is as close to the chALL cluster as the CB *MLL-Af4* samples.

We have amended our discussion to point out only that *CRISPR*^{MLL-AF4+} ALL clusters with *MLL-AF4* infant-ALL patients (lines 196-197):

“We hypothesized that this model may represent a neonatally-derived (non-fetal) ALL to which our model could be compared.”

and lines 202-204:

“Clustering analysis based on this core fetal-specific infant-ALL gene list showed that *CRISPR*^{MLL-AF4+} ALL clustered with *MLL-AF4* infant-ALL, whereas both *MLL-AF4* childhood-ALL and CB *MLL-Af4* ALL formed their own, separate clusters (Figure 5a)”.

Based on this, we would interpret that *CRISPR*^{MLL-AF4+} ALL is an accurate model of poor prognosis infant-ALL without drawing any conclusions about the CB *MLL-Af4* ALL model being more similar to childhood-ALL, discussed in lines 247-249:

“Our results confirm that a human fetal cell context is permissive to give rise to an ALL that recapitulates key phenotypic and molecular features of poor prognosis *MLL-AF4* infant-ALL”.

10. While there is a lot of evidence that the fetal cell of origin is an important contributing factor to the infant-ALL phenotype, I don't think this was convincingly shown in this study. The fact that the ^{CRISPR}*MLL-AF4+* samples clustered with the *MLL-AF4* infant-ALL samples and away from the CB *MLL-Af4* samples is not surprising. It could easily be explained by the fact that the authors managed to induce a true t(4;11) translocation, including expression of the reciprocal *AF4-MLL* reciprocal fusion, that resembles the human disease situation much more closely than viral overexpression of a human:mouse *MLL-Af4* chimaeric fusion. To demonstrate that it is entirely due to the fetal context, they would have to demonstrate that CRISPR-induced t(4;11) in CB cells produces a disease that clusters with the childhood-ALL samples. The fact that there are subtypes even within the infant *MLL-AF4* ALL patients, some of which resemble childhood cases, suggests that it may be more complicated than that, and that the specific progenitor subtype may be equally important. This may also explain why the authors obtained both proB as well as preB phenotypes from fetal cells – they may have induced the translocation in different progenitor types within the broad CD34+ population. This should be acknowledged and discussed.

We thank the reviewer for raising this point. Although we feel that our data as a whole support the fact that a fetal cell context is likely to play a key role in the accuracy of our model, we agree there could be several reasons for the differences we demonstrate from a previously published CB model, including the exact nature of translocations and fusion proteins generated. The similarity of our fetal model to infant-ALL could potentially be due to ^{CRISPR}*MLL-AF4+* ALL having a bona fide translocation found in infant-ALL patients, making it a more accurate model of *MLL-AF4* ALL in general. However, it is perhaps worth mentioning that Figure 5a, in which ^{CRISPR}*MLL-AF4+* ALL clusters with *MLL-AF4* infant-ALL, is based on a fetal-specific infant-ALL gene signature rather than an *MLL-AF4* ALL signature which would be present in both infant-ALL and childhood-ALL clinical samples with t(4;11)/*MLL-AF4* translocations. In agreement with the reviewer's comments that the published CB model (Lin et al.) has not been generated using the same approaches used by us, we have removed any discussion suggesting that ^{CRISPR}*MLL-AF4+* ALL is more similar to *MLL-AF4* infant-ALL than CB *MLL-Af4+* ALL due to its fetal origin.

In our revised manuscript, we mention only that *MLL-AF4* binding at the promoters of fetal-specific infant-ALL genes suggests that, in our model, *MLL-AF4* likely cooperates with this fetal-specific gene expression program to initiate ^{CRISPR}*MLL-AF4+* ALL.

We also agree that the heterogeneity within *MLL-AF4* infant-ALL, which we have also demonstrated in our model, may be driven by the specific hematopoietic progenitor subtype that gets transformed in different patients, and have now made this clearer in the discussion (lines 250-265):

“We targeted the t(4;11)/*MLL-AF4* translocation to CD34+ FL cells, which represent a mixture of different HSPC types. The immunophenotypic heterogeneity we observed among primary ^{CRISPR}*MLL-AF4+* mice, with 2/3 showing a proB and 1/3 showing a preB immunophenotype, may be a consequence of the translocation occurring in different progenitor cell types. Interestingly however, no other significant differences were observed between proB and preB ^{CRISPR}*MLL-AF4+* ALL. Firstly, no clinico-pathological differences were observed, which may suggest that it is the shared fetal characteristics, more so than a cell-type-specific context, that drive the aggressive phenotypic features of infant-ALL, such as treatment-resistance and CNS disease. Secondly, all ^{CRISPR}*MLL-AF4+* ALLs represented the *HOXA*^{lo} subset of *MLL-AF4* infant-ALL. While this may draw an interesting parallel with the higher frequency of the *HOXA*^{lo} subset observed in *MLL-r* infant-ALL patients, we cannot draw conclusions from these data about the specific cell of origin of infant-ALL and/or the

“A novel human fetal liver-derived model reveals that MLL-AF4 drives a distinct fetal gene expression program in infant-ALL”

drivers of the $HOXA^{lo}/HOXA^{hi}$ molecular profiles. It will be interesting in the future to target the t(4;11)/MLL-AF4 translocation to specific fetal HSPC subsets to ask whether leukemic transformation and *HOXA* status is determined by gestational age, hematopoietic site or progenitor cell type.”

REFERENCES

- 1 Pieters, R. *et al.* In vitro drug sensitivity of cells from children with leukemia using the MTT assay with improved culture conditions. *Blood* **76**, 2327-2336 (1990).
- 2 Pieters, R. *et al.* Relation between age, immunophenotype and in vitro drug resistance in 395 children with acute lymphoblastic leukemia--implications for treatment of infants. *Leukemia* **12**, 1344-1348, doi:10.1038/sj.leu.2401129 (1998).
- 3 Ramakers-van Woerden, N. L. *et al.* In vitro drug-resistance profile in infant acute lymphoblastic leukemia in relation to age, MLL rearrangements and immunophenotype. *Leukemia* **18**, 521-529, doi:10.1038/sj.leu.2403253 (2004).

REVIEWER COMMENTS

Reviewer #1 (Remarks to the Author):

The authors have nicely and comprehensively handled my suggestions, I have no further suggestions.

Reviewer #2 (Remarks to the Author):

The authors have done a great job at answering my queries. There are just a few issues that have arisen as a consequence of the revisions which need clarifying.

1. In lines 86/87, it says: "...we found 72 genes that were significantly upregulated in both normal FL HSPCs and MLL-AF4 infant-ALL..." – this sounds as if those 72 genes were upregulated in all normal FL HSPC populations and the ALL samples, which is not the case. To be more precise, the sentence should be reworded to: "...we found 72 genes that were significantly upregulated in at least one normal FL HSPC population and MLL-AF4 infant-ALL..."
2. In Supp. Table 3, rows 10 & 11, in vivo drug chemotherapy drug treatments are listed; however, the text in lines 137-144 only describes in vitro experiments. Those two lines should either be removed from the table or the in vivo results discussed.
3. The new plot in Fig. 4c is a nice addition and was generated from the data in Supp Table 4, which came from a 3-way statistical comparison between HOXA^{lo} MLL-AF4 infant-ALL, HOXA^{hi} MLL-AF4 infant-ALL and HOXA^{hi} MLL-AF4 childhood-ALL. For each gene in Supp. Table 4, however, only one logFC, PValue and FDR value are shown, so it is impossible to know which specific comparison these values come from.

Reviewer #3 (Remarks to the Author):

I did not review the first version of this manuscript, but at the request of the editors of Nature Communications, I have examined both the original and revised versions of this study, the previous reviewers' comments, and the authors' rebuttal.

Overall, the authors have taken the previous reviewers' comments seriously and have performed relevant modifications in the manuscript to address these comments.

For example, to address the comments of Reviewer 1, the authors have added new data in which drug (prednisolone and L-asparaginase) sensitivity of ALL blasts from PDX models, as well as the SEM and KOPN8 cell lines, was compared with CRISPR_MLL-AF4+ cells. To address the comments raised by Reviewer 2, the authors have performed a 3-way statistical comparison between HOXA low and high MLL-AF4 childhood-ALL. The revised text addresses all reviewers' suggestions satisfactorily.

I agree with Reviewers 1 and 2 about the important contribution of the generation of the first fully humanized mouse model of MLL-AF4+ infant ALL that recapitulates the human disease. I do, however, have some concerns:

- 1.- The authors confirm the expression of both MLL-AF4 and AF4-MLL in CRISPR_MLL-AF4 in vitro and in vivo models by RT-qPCR. FISH analysis has only been developed in the in vivo models. Surprisingly, the authors did not incorporate a complete characterization of CRISPR_MLL-AF4 cells used in the in vitro and in vivo models. Karyotype, FISH, and aCGH or NGS analysis must be done to characterize the genome of the cells. It is known that the use of two sgRNAs at the same time can produce genomic rearrangements and are associated with off-target effects. This analysis could also help to quantify the percentage of cells harboring the chromosomal translocation and confirm the

presence of the two derivative chromosomes in all the cells, the loss of any derivative chromosome, or the presence of other rearranged chromosomes.

2.- It is not clear to me if the authors have used cryopreserved CD34+ cells from 6 different donors in the 3 in vitro and 3 in vivo edited samples used in the study. This should be explained clearly in the manuscript. This is important to understand whether the genomic background of the donors could have a role in the development of the infant ALL. In this regard, I found quite low the number of in vitro and in vivo replicas.

3.- The authors hypothesized that the humanized mouse model of MLL-AF4 ALL previously published with a chimeric MLL-Af4 fusion gene in CB HSPCs may represent a non-fetal ALL to which the CRISPR_MLL-AF4 model could be compared. Those two models are completely different (i.e., overexpression versus generation of chromosomal translocations, murine Af4 partner gene), and cannot be compared. The authors should have recreated the t(4;11) using CRISPR in CB-derived cells to examine the fetal and post-natal gene expression programs. This would be a great advance in the understanding of the origin of this infant leukemia and generate a great set of comparable data.

4.- Another important aspect is to clarify, firstly, why the authors use an MS5 co-culture system as a standard to evaluate the generation of a bona fide model of MLL-AF4 translocation and finally why did not use a liquid culture model that could demonstrate the immortalization (at least partial) of the leukemic stem cells generated with this approach.

5.- Please cite the following publications along with reference 48 when describing the CRISPR-Cas9 genome editing strategy:

- Choi, P., Meyerson, M. Targeted genomic rearrangements using CRISPR/Cas technology. *Nat Commun* 5, 3728 (2014). <https://doi.org/10.1038/ncomms4728>

- Torres, R., Martin, M., Garcia, A. et al. Engineering human tumour-associated chromosomal translocations with the RNA-guided CRISPR–Cas9 system. *Nat Commun* 5, 3964 (2014). <https://doi.org/10.1038/ncomms4964>

We would like to thank the reviewers for their additional constructive comments. We have generated some new data and re-written parts of the manuscript to address these comments. A specific point-by-point response is included below.

Summary of changes to figures and tables:

Figure 4c: New UMAP based on 765 differentially expressed genes between *HOXA*^{lo} infant-ALL, *HOXA*^{hi} infant-ALL and *HOXA*^{hi} childhood-ALL. Revised after correction of RNA-seq analysis. Please see reviewer 2 comment 3.

Supplementary Figure 3b: New figure. *MLL-AF4/t(4;11)* FISH analysis of *CRISPR**MLL-AF4+* cells generated *in vitro*

Supplementary Figure 3c: Previously Supplementary Figure 3b.

Supplementary Figure 3d: Previously Supplementary Figure 3c.

Supplementary Figure 4c: Additional *MLL-AF4/t(4;11)* FISH in *CRISPR**MLL-AF4+* ALL (from primary recipient mouse spleen).

Supplementary Figure 4d: Previously Supplementary Figure 4c.

Supplementary Figure 4e: Summary of indels present in the wild-type allele of *MLL* in *CRISPR**MLL-AF4+* ALL from 3 primary recipient mice (preB and proB ALL, from donors 1 and 2) compared to matched, unedited FL cells (donor 1 and 2).

Supplementary Figure 4f: FISH analysis of the other four most common *MLL* fusion genes in *CRISPR**MLL-AF4+* ALL from primary recipient mice.

Supplementary Figure 4g: Karyotype analysis of *CRISPR**MLL-AF4+* ALL from primary recipient mice.

Supplementary Figure 4h: Previously Supplementary Figure 4d and 4e.

Supplementary Figure 4i: Previously Supplementary Figure 4f.

Supplementary Table 3: Summary of genomic analyses in *CRISPR**MLL-AF4+* ALL (FISH, karyotyping and Sanger sequencing).

Supplementary Table 4: Previously Supplementary Table 3. Label changes (please see reviewer 2 comment 2). “Karyotype” and “*MLL-AF4/t(4;11)* positive by FISH” rows added.

Supplementary Table 5: Previously Supplementary Table 4. Revised analysis of *HOXA*^{lo} infant-ALL, *HOXA*^{hi} infant-ALL and *HOXA*^{hi} childhood-ALL RNA-seq data, showing all LogFC values for each group. Please see reviewer 2 comment 3.

Supplementary Table 6: Previously Supplementary Table 5.

Supplementary Table 7: Previously Supplementary Table 6.

REVIEWER COMMENTS

Reviewer #1 (Remarks to the Author):

The authors have nicely and comprehensively handled my suggestions, I have no further suggestions.

Thank you very much for your previous suggestions which we think have improved the paper.

Reviewer #2 (Remarks to the Author):

The authors have done a great job at answering my queries. There are just a few issues that have arisen as a consequence of the revisions which need clarifying.

1. In lines 86/87, it says: “...we found 72 genes that were significantly upregulated in both normal FL HSPCs and *MLL-AF4* infant-ALL...” – this sounds as if those 72 genes were upregulated in all normal FL HSPC populations and the ALL samples, which is not the case. To be more precise, the sentence should be reworded to: “...we found 72 genes that were significantly upregulated in at least one normal FL HSPC population and *MLL-AF4* infant-ALL...”

Thank you for pointing this out. This wording is clearer and we have made the suggested change to the text (page 6, lines 86-87).

2. In Supp. Table 3, rows 10 & 11, *in vivo* drug chemotherapy drug treatments are listed; however, the text in lines 137-144 only describes *in vitro* experiments. Those two lines should either be removed from the table or the *in vivo* results discussed.

We apologize for the confusing wording. These rows represent whether the ALL from *in vivo* models were resistant to prednisolone or L-asparaginase *in vitro* (i.e. leukemic blasts were taken from the *in vivo* primary xenograft model and tested for drug resistance *in vitro*, as has previously been published for primary patient samples^{1,2}). To make this clearer, we have renamed these rows “Pred resistant *in vitro* (LC50>100ug/ml)” and “L-asp resistant *in vitro* (LC50>0.1IU/ml)”.

3. The new plot in Fig. 4c is a nice addition and was generated from the data in Supp Table 4, which came from a 3-way statistical comparison between *HOXA*^{lo} *MLL-AF4* infant-ALL, *HOXA*^{hi} *MLL-AF4* infant-ALL and *HOXA*^{hi} *MLL-AF4* childhood-ALL. For each gene in Supp. Table 4, however, only one logFC, p value and FDR value are shown, so it is impossible to know which specific comparison these values come from.

Thank you for raising this point. We have rerun this analysis in the following manner to output all relevant statistics:

Using the edgeR package, a design matrix was built to include *HOXA*^{hi} *MLL-AF4* childhood-ALL, *HOXA*^{hi} *MLL-AF4* infant-ALL and *HOXA*^{lo} *MLL-AF4* infant ALL as 3 separate conditions. We then carried out an ANOVA-style comparison between all conditions as follows:

```
fit<-glmQLFit(counts, design)
qlf<-glmQLFTest(fit, coef=2:ncol(design))
```

By taking the topTags from this 3-way comparison and filtering for FDR<0.05, we identified “marker”-type genes that differentiate the 3 conditions in the design matrix from one another.

After rerunning the analysis, we identified a slightly lower number of significantly differentially expressed genes (765 genes). The analysis used has been clarified in the updated methods section (page 21, lines 481-486). We now show these data in Supplementary Table 5, with all LogFC values included. UMAP analysis including the patient samples and *CRISPR*^{MLL-AF4+} based on these 765 genes gave the same results, whereby *CRISPR*^{MLL-AF4+} samples cluster with *HOXA*^{lo} *MLL-AF4* infant-ALL and far from all *HOXA*^{hi} samples (new Figure 4c).

Reviewer #3 (Remarks to the Author):

I did not review the first version of this manuscript, but at the request of the editors of Nature Communications, I have examined both the original and revised versions of this study, the previous reviewers' comments, and the authors' rebuttal.

Overall, the authors have taken the previous reviewers' comments seriously and have performed relevant modifications in the manuscript to address these comments.

For example, to address the comments of Reviewer 1, the authors have added new data in which drug (prednisolone and L-asparaginase) sensitivity of ALL blasts from PDX models, as well as the SEM and KOPN8 cell lines, was compared with *CRISPR*^{MLL-AF4+} cells. To address the comments raised by Reviewer 2, the authors have performed a 3-way statistical comparison between *HOXA* low and high *MLL-AF4* childhood-ALL. The revised text addresses all reviewers' suggestions satisfactorily.

I agree with Reviewers 1 and 2 about the important contribution of the generation of the first fully humanized mouse model of *MLL-AF4+* infant ALL that recapitulates the human disease. I do, however, have some concerns:

1.- The authors confirm the expression of both *MLL-AF4* and *AF4-MLL* in *CRISPR*^{MLL-AF4} *in vitro* and *in vivo* models by RT-qPCR. FISH analysis has only been developed in the *in vivo* models. Surprisingly, the authors did not incorporate a complete characterization of *CRISPR*^{MLL-AF4+} cells used in the *in vitro* and *in vivo* models. Karyotype, FISH, and aCGH or NGS analysis must be done to characterize the genome of the cells. It is known that the use of two sgRNAs at the same time can produce genomic rearrangements and are associated with off-target effects. This analysis could also help to quantify the percentage of cells harboring the chromosomal translocation and confirm the presence of the two derivative chromosomes in all the cells, the loss of any derivative chromosome, or the presence of other rearranged chromosomes.

We thank the reviewer for raising these important points.

First, regarding the characterization of the *CRISPR* *MLL-AF4*+ cells used in the *in vitro* and *in vivo* assays, we have now included more *MLL-AF4* FISH data for both *in vitro* and *in vivo* *CRISPR* *MLL-AF4*+ cells. Specifically, we have added the percentage of cells positive for *MLL-AF4* and/or *AF4-MLL* scored from an analysis of 200-2,000 cells. This showed that 81-99% of *in vitro* *CRISPR* *MLL-AF4*+ cells (n=4) (Reviewer table 1) and 87-99% of *in vivo* *CRISPR* *MLL-AF4*+ cells (n=3) (Reviewer Table 2) harbor the translocation. These data are now included in Supplementary Figure 3b for *in vitro* assays, Supplementary table 4 and Supplementary Figure 4c for *in vivo* assays, and in results (page 7, lines 109-111 and page 8, lines 137-138). We apologize for having omitted this detailed information in the previous version of the manuscript.

Reviewer Table 1

In vitro		MLL-AF4 FISH
FL donor 4	rep1	Fusion detected in 97%
	rep2	Fusion detected in 80.5%
FL donor 5	rep1	Fusion detected in 97%
	rep2	Fusion detected in 98.5%

Second, to exclude the generation of other *MLL* fusion genes in our model, we have now performed FISH analysis for the other four most common fusion partners of *MLL* (*AF6*, *AF9*, *ENL*, *AF10*). This confirmed that *MLL* has not translocated to any of these *MLL* partner genes and also allowed us to confirm that both derivative chromosomes are present in all *CRISPR* *MLL-AF4*+ cells (these data have been added as Supplementary Figure 4f, and described in results, page 8 lines 141-142) (Reviewer Table 2). The presence of both derivative fusion genes and their expression was also definitively confirmed by our qPCR results showing expression of *MLL-AF4* and *AF4-MLL* transcripts in *CRISPR* *MLL-AF4*+ cells from *in vivo* assays (Supplementary Fig 4b) and at the genomic level by the Sanger sequencing results (Supplementary Fig 4d)

Reviewer Table 2

In vivo		FISH				
		MLL-AF4	MLL-ENL	MLL-AF10	MLL-AF9	MLL-AF6
FL donor 1	PreB ALL	Fusion detected in 87%	no fusion, extra MLL signal (81%)	no fusion, extra MLL signal (77%)	no fusion, extra MLL signal (76%)	no fusion, extra MLL signal (84%)
FL donor 2	ProB ALL	Fusion detected in 97%	no fusion, extra MLL signal (84%)	no fusion, extra MLL signal (88%)	no fusion, extra MLL signal (88%)	no fusion, extra MLL signal (91%)
	ProB ALL	Fusion detected in 98.5%	no fusion, extra MLL signal (97%)	no fusion, extra MLL signal (95%)	no fusion, extra MLL signal (87%)	no fusion, extra MLL signal (98%)

Thirdly, to further test the possibility that *MLL* and/or *AF4* might have translocated with any other partner gene, and to look for any other large structural abnormalities (e.g. translocations other than *MLL-AF4*) caused by off-target editing, we have carried out karyotyping analysis on all primary ^{CRISPR}*MLL-AF4*+ ALL samples. This has confirmed that there are no major structural changes or translocations post editing in the transformed cells other than t(4;11) and these data have been added to results (page 8, lines 142-144, Supplementary Fig 4g, Supplementary Table 4). (Note: one FL donor had a constitutional der(14;21) that was confirmed in non-edited FL cells, and is not expected to be pathogenic). This has been noted in the methods section (page 17, lines 382-385) (Reviewer Fig 1)

Reviewer Fig 1

Unedited FL cells (donor 2)

^{CRISPR}*MLL-AF4*+ ALL (donor 2)

Reviewer Figure 1: Karyotyping of unedited FL cells from donor 2 on the left showing der(14;21), marked by arrows; karyotyping of ^{CRISPR}*MLL-AF4*+ ALL cells derived from donor 2 primary xenografts showing the same constitutional der(14;21) marked by arrows and CRISPR-Cas9 mediated t(4;11), highlighted by red boxes

Finally, as an additional check, we examined the in silico-predicted off-target sites (predicted using Synthego’s CRISPR Guide Verification tool; <https://design.synthego.com/#/validate>) with less than 4 mismatches in the recognition sequence. No predicted off-target site had less than 3 mismatches (suggesting high specificity). Of those targets with 3 mismatches, all but one (*KCNQ2*, exon 17) were found in introns and >1kb away from flanking exons. There were a total of 9 such sites, 7 off-target genes for *MLL*-sgRNA and 2 off-target genes for *AF4*-sgRNA. We have sequenced all 9 gene regions for our xenograft derived ^{CRISPR}*MLL-AF4*+ ALL cells (n=3) and run the sequencing data through Synthego’s ICE analysis too (<https://ice.synthego.com>), which compared sequencing data from the edited sample (^{CRISPR}*MLL-AF4*+ ALL) to an unedited sample (matched, unedited cells from the same FL donor). We found that no indels were present at these loci. These data are described in results (page 8, lines 144-147) and tabulated in Supplementary Table 3.

Representative output from the Synthego ICE analysis tool is included below (Reviewer Figure 2). 0% editing efficiency means that 0% of the reads in the edited sample contain indels compared to the matched, unedited control FL cells. In the manuscript (Supplementary Table 3), we have reported this as the edited sample being 100% similar to the matched, unedited control FL cells.

Reviewer Figure 2

(a)

(b)

(c)

Reviewer Figure 2: Off-target editing analysis at (a) *KDM6B* (*MLL*-sgRNA potential target), (b) *KCNQ2* (*AF4*-sgRNA potential target) and (c) the wild-type allele of *MLL*. Edited sample = primary proB *CRISPR*^{MLL-AF4+} ALL (from donor 2), control sample = unedited FL cells (donor 2). “Indel %” shows percentage of reads in edited sample that have indels compared to control sample. “Model Fit (R²)” reflects the quality of the sequencing data used in the analysis (over 0.9 is recommended by Synthego website). (c) shows that the *CRISPR*^{MLL-AF4+} ALL clone has a 3bp deletion at the *MLL* cut site.

2.- It is not clear to me if the authors have used cryopreserved CD34+ cells from 6 different donors in the 3 *in vitro* and 3 *in vivo* edited samples used in the study. This should be explained clearly in the manuscript. This is important to understand whether the genomic background of the donors could have a role in the development of the infant ALL. In this regard, I found quite low the number of *in vitro* and *in vivo* replicas.

We apologize that the information about the biological samples used for *in vitro* and *in vivo* assays was not clear. *In vitro* assays were performed using 3 biological donors (both ^{CRISPR}MLL-AF4+ and control cells from donors 1-3). *In vivo* assays were performed using the same 3 donors plus one additional donor (^{CRISPR}MLL-AF4+ ALL from donors 1 and 2, control xenografts from donors 1, 2, 3 and 6). We have since performed two more *in vitro* experiments from new donors (donors 4 and 5) and MLL-AF4 FISH data from these donors are now included in the manuscript. We have an additional *in vivo* experiment from donor 7, where all 4 transplanted mice developed ALL in primary xenograft assays, however this data has not been included as secondary and tertiary xenograft assays are not complete. We have now included donor information throughout the manuscript. When sample numbers are given, the donors are specified (e.g. (n=3; donors 1-3) for *in vitro* and (^{CRISPR}MLL-AF4+ n=3: donors 1 and 2; control n=5: donors 1, 2, 3 and 6) for *in vivo*. Although our replicates are low in number, the transformation rate is 100%, as every biological FL sample *in vitro* and/or *in vivo* has transformed. Available baseline karyotype data of the donors were all normal, except for donor 2. A table summarizing all donors is included below (Reviewer table 3):

Reviewer Table 3:

Fetal liver	Gestational age	Original Karyotype	in vitro transformation	in vivo ALL
donor 1	13 pcw	46, XY	yes	yes (1/1 mouse)
donor 2	13 pcw	45, XY,der(14;21)(q10;q10)	yes	yes (2/2 mice)
donor 3	15 pcw	46, XY	yes	controls only
donor 4	17 pcw	failed	yes	ND
donor 5	17 pcw	46, XX	yes	ND
donor 6	14 pcw	46, XY	ND	controls only
donor 7 (not included in manuscript)	13 pcw	46, XX	ND	yes (4/4 mice)

3.- The authors hypothesized that the humanized mouse model of MLL-AF4 ALL previously published with a chimeric MLL-Af4 fusion gene in CB HSPCs may represent a non-fetal ALL to which the ^{CRISPR}MLL-AF4 model could be compared. Those two models are completely different (i.e., overexpression versus generation of chromosomal translocations, murine Af4 partner gene), and cannot be compared. The authors should have recreated the t(4;11) using CRISPR in CB-derived cells to examine the fetal and post-natal gene expression programs. This would be a great advance in the understanding of the origin of this infant leukemia and generate a great set of comparable data.

We agree that it would be interesting to compare ^{CRISPR}MLL-AF4+ FL and cord blood (CB) but the aim of our study was to create an MLL-AF4 model in fetal cells and to determine if we could identify a "fetal" gene signature in unedited fetal HSPC which was preserved both in our ^{CRISPR}MLL-AF4+ FL model and in primary ALL samples. We felt it would be relevant to draw attention in the results/discussion to the previous best model of MLL-Af4 ALL in CB cells but we acknowledge that the CB model in Lin et al. might be different from our FL model for many reasons, including the exact nature of translocations and fusion proteins generated, and we

have rephrased the wording of the results/discussion accordingly in the previous revision of the manuscript:

From previous response:

"...Based on this, we would interpret that ^{CRISPR}MLL-AF4+ ALL is an accurate model of poor prognosis infant-ALL without drawing any conclusions about the CB MLL-Af4 ALL model being more similar to childhood-ALL, discussed in lines 246-248 [now 263-265]:

“Our results confirm that a human fetal cell context is permissive to give rise to an ALL that recapitulates key phenotypic and molecular features of poor prognosis MLL-AF4 infant-ALL”.

4.- Another important aspect is to clarify, firstly, why the authors use an MS5 co-culture system as a standard to evaluate the generation of a bona fide model of MLL-AF4 translocation and finally why did not use a liquid culture model that could demonstrate the immortalization (at least partial) of the leukemic stem cells generated with this approach.

An MS-5 stromal co-culture system was used to allow edited cells to grow out because this system is permissive to multi-lineage (myeloid, NK and B cell) output, while in our experience liquid culture systems of human cells are strongly biased towards myeloid outputs and although suitable for the culture of AML blasts, they rarely support B-ALL blasts. In keeping with this, primary patient-derived B-ALLs have only been reported to survive for extended periods *in vitro* when co-cultured with a supportive stromal layer³. As MLL-AF4 ALL is almost invariably a B-ALL, and because our aim was to recapitulate infant MLL-AF4+ B-ALL, we opted for a culture system that is permissive to B cell output and will allow optimum outgrowth of a transformed B-ALL clone.

5.- Please cite the following publications along with reference 48 when describing the CRISPR-Cas9 genome editing strategy:

- Choi, P., Meyerson, M. Targeted genomic rearrangements using CRISPR/Cas technology. *Nat Commun* 5, 3728 (2014). <https://doi.org/10.1038/ncomms4728>

- Torres, R., Martin, M., Garcia, A. et al. Engineering human tumour-associated chromosomal translocations with the RNA-guided CRISPR–Cas9 system. *Nat Commun* 5, 3964 (2014). <https://doi.org/10.1038/ncomms4964>

We thank the reviewer for suggesting these citations, and apologize for their omission; they now have been included in the appropriate locations throughout the manuscript.

References:

- 1 Pieters, R. *et al.* Relation of cellular drug resistance to long-term clinical outcome in childhood acute lymphoblastic leukaemia. *Lancet* **338**, 399-403, doi:10.1016/0140-6736(91)91029-t (1991).
- 2 Pieters, R., Kaspers, G. J., Klumper, E. & Veerman, A. J. Clinical relevance of in vitro drug resistance testing in childhood acute lymphoblastic leukemia: the state of the art. *Med Pediatr Oncol* **22**, 299-308, doi:10.1002/mpo.2950220502 (1994).
- 3 Pal, D. *et al.* Long-term in vitro maintenance of clonal abundance and leukaemia-initiating potential in acute lymphoblastic leukaemia. *Leukemia* **30**, 1691-1700, doi:10.1038/leu.2016.79 (2016).

REVIEWER COMMENTS

Reviewer #2 (Remarks to the Author):

The authors have answered all of my previous comments. The only remaining thing I would like the authors to do is to remove two novelty claims from the Discussion. On page 12, line 263, they claim that they have created “the first bona fide MLL-AF4 infant-ALL model”, which is no longer the case, as Malouf et al. (PMID: 34111240) have in the meantime published a study which also reports the generation of a representative MLL-AF4 pro-B ALL model that recapitulates many features of the infant disease including CNS infiltration. I would suggest that the sentence is changed to “we have created a faithful MLL-AF4 infant-ALL model”. For the same reason, the phrase “previously lacking” should be removed from page 13, line 288, as the new Malouf et al. model mentioned above is also a useful preclinical model in which CNS disease can be studied.

Reviewer #3 (Remarks to the Author):

The authors have nicely and comprehensively handled my suggestions. I have no further comments.

LIST OF REVISIONS TO FIGURES AND TABLES

Figure 1d. – Schematic fetus and adult redrawn

Figure 3b – Scale bar added

Figure 5c – Bar plots changed to box-and-whisker plots

Supplementary Figure 3b – Scale bars added

Supplementary Figure 4f – Scale bars added

Supplementary Figure 5d – Minor adjustments to CD34 % values for 2 datapoints

Supplementary Figure 6a – Sorting strategy added

Supplementary Figure 6b – Previously Supplementary Figure 6a

Supplementary Figure 6c – Previously Supplementary Figure 6b

Supplementary Figure 6d – Previously Supplementary Figure 6c

Supplementary Table 1 – Previously Supplementary Table 3

Supplementary Table 2 – Previously Supplementary Table 4

Supplementary Table 3-5 – Previously Supplementary Table 7

Supplementary Data 1 – Previously Supplementary Table 1

Supplementary Data 2 – Previously Supplementary Table 2

Supplementary Data 3 – Previously Supplementary Table 5

Supplementary Data 4 – Previously Supplementary Table 6

Source data file now included

REVIEWERS' COMMENTS

Reviewer #2 (Remarks to the Author):

The authors have answered all of my previous comments. The only remaining thing I would like the authors to do is to remove two novelty claims from the Discussion. On page 12, line 263, they claim that they have created “the first bona fide *MLL-AF4* infant-ALL model”, which is no longer the case, as Malouf et al. (PMID: 34111240) have in the meantime published a study which also reports the generation of a representative *MLL-AF4* pro-B ALL model that recapitulates many features of the infant disease including CNS infiltration. I would suggest that the sentence is changed to “we have created a faithful *MLL-AF4* infant-ALL model”. For the same reason, the phrase “previously lacking” should be removed from page 13, line 288, as the new Malouf et al. model mentioned above is also a useful preclinical model in which CNS disease can be studied.

We thank the reviewer for these suggestions and all previous constructive comments. Page 12, line 263 now reads:

“we have created a faithful humanized *MLL-AF4* infant-ALL model”

On page 13, line 288, we have removed “previously lacking”. It now reads:

“*CRISPR* *MLL-AF4+* ALL provides a preclinical model for translational studies that specifically recapitulates poor prognosis infant-ALL.”

Reviewer #3 (Remarks to the Author):

The authors have nicely and comprehensively handled my suggestions. I have no further comments.

We thank the reviewer for their suggestions.